# Molecular characterization and population genetics of *Theileria parva* in Burundi's unvaccinated cattle: Towards the introduction of East Coast fever vaccine

David Kalenzi Atuhaire[1☉], Walter Muleya[2☉]*, Victor Mbao[3], Joseph Niyongabo[4‡], Lionel Nyabongo[4‡], Deogratias Nsanganiyumwami[4‡], Jeremy Salt[5], Boniface Namangala[6], Antony Jim Musoke[7]

1 Centre for Ticks and Tick-Borne Diseases, Lilongwe, Malawi, 2 Department of Biomedical Sciences, School of Veterinary Medicine, University of Zambia, Lusaka, Zambia, 3 International Development Research Centre, Eastern and Southern Africa Regional Office, Nairobi, Kenya, 4 National Veterinary Research Laboratory, Directorate of Animal Health, Bujumbura, Burundi, 5 Global Alliance for Livestock Veterinary Medicines, Pentlands Science Park, Bush Loan, Penicuik Edinburgh, Scotland, 6 Department of Paraclinical Studies, School of Veterinary Medicine, University of Zambia, Lusaka, Zambia, 7 LMK Medical laboratories and Consultancies, Kampala, Uganda

☉ These authors contributed equally to this work.
‡ These authors also contributed equally to this work.
* muleyawalter@gmail.com

**Data Availability Statement:** The sequences emanating from this study were deposited in the

## Abstract

*Theileria parva* (*T. parva*) is a protozoan parasite that causes East Coast fever (ECF). The disease is endemic in Burundi and is a major constraint to livestock development. In this study, the parasite prevalence in cattle in six regions namely; Northern, Southern, Eastern, Western, Central and North Eastern was estimated. Furthermore, the sequence diversity of p67, Tp1 and Tp2 genes was assessed coupled with the population genetic structure of *T. parva* using five satellite markers. The prevalence of ECF was 30% (332/1109) on microscopy, 60% (860/1431) on ELISA and 79% (158/200) on p104 gene PCR. Phylogenetic analysis of p67 gene revealed that only allele 1 was present in the field samples. Furthermore, phylogenetic analysis of Tp1 and Tp2 showed that the majority of samples clustered with Muguga, Kiambu and Serengeti and shared similar epitopes. On the other hand, genetic analysis revealed that field samples shared only two alleles with Muguga Cocktail. The populations from the different regions indicated low genetic differentiation ($F_{ST}$ = 0.047) coupled with linkage disequilibrium and non-panmixia. A low to moderate genetic differentiation ($F_{ST}$ = 0.065) was also observed between samples and Muguga cocktail. In conclusion, the data presented revealed the presence of a parasite population that shared similar epitopes with Muguga Cocktail and was moderately genetically differentiated from it. Thus, use of Muguga Cocktail vaccine in Burundi is likely to confer protection against *T. parva* in field challenge trials.

Gene Bank with accession numbers LC593820 to LC594067 for Tp1 and 2 and LC594555to LC 594610 for p67. All relevant data are within the manuscript and its Supporting information files.

**Funding:** Financial support for this work was within the framework of the project on the characterization and population genetics of Theileria parva strains in eastern, central and southern Africa, grant number UOZ-R34A0548A3 received from the Global alliance for livestock medicines (GALVmed). The funders had no role in study design, data collection and analysis, decision to publish, or preparation of the manuscript.

**Competing interests:** The authors have declared that no competing interests exist.

**Abbreviations:** CTL, Cytotoxic T Lymphocytes; CTTBD, Centre for Ticks and Tick-Borne Diseases; ECF, East Coast Fever; LD, Linkage Disequilibrium; LE, Linkage Equilibrium; MC, Muguga Cocktail; MLG, Multi-Locus Genotype; PCA, Principal Component Analysis; PCR, Polymerase Chain Reaction.

# Introduction

*Theileria parva* is an obligate intracellular protozoan parasite transmitted by the three-host tick known as *Rhipicephalus appendiculatus*. It causes East Coast Fever (ECF), a devastating disease associated with high mortality rates in susceptible cattle populations [1]. ECF is prevalent wherever the vector exists in the Eastern, Central and Southern parts of Africa and has been reported in 11 countries; Burundi, Rwanda, Uganda, Kenya, South Sudan, Tanzania, Malawi, Democratic Republic of Congo, Zimbabwe, Mozambique and Zambia [2]. Between 2003 and 2004 ECF was reported for the first time in the Comoros islands and this was attributed to importation of vaccinated cattle from Tanzania [3]. Naïve ticks fed on immunised cattle and subsequently transmitted the infection to susceptible local cattle population [3]. Since its detection, the disease is now ranked the most important disease of cattle in Comoros [4].

The control of ECF is usually by the use of acaricide to prevent tick challenge or by immunisation using the infection and treatment method (ITM). Drugs are available for treatment of clinically ill cases but they are extremely expensive. Kiltz and Humke [5] reported the successful use of Halofuginone lactate in the control of ECF under field conditions. The ITM involves administering a lethal dose of *T. parva* in combination with a long-acting formulation of oxytetracycline [6]. One of the widely used ITM in East Africa is the Muguga Cocktail (MC) that comprises three *T. parva* parasite stocks; Muguga and Kiambu 5 isolated from cattle in Kenya and Serengeti-transformed, the buffalo derived isolate from Tanzania [7]. Other characterised and routinely used ECF vaccine stocks are Chitongo and Katete isolates in Zambia [8], Marikebuni in Kenya [9] and Boleni isolate in Zimbabwe [10]. The Nyakizu and Gikongoro isolates in Rwanda [11] and the Gatumba, Gitega and Ngozi isolates in Burundi [12] have been characterised and only used on an experimental basis. The immunity acquired from ITM may last for many years and resists homologous challenge, but not necessarily to all strains encountered in the field. Failure to provide full protection has been attributed to the existence of variability in the parasite population. Cattle that recover from the disease become carriers, thus increasing the infection rate in ticks and consequently the inoculation rate of other cattle [13]. The wider application of the ITM vaccine in most endemic areas has been hampered by the growing concern over the possibility of introduction of vaccine strains following immunization against ECF. This has limited the extensive use of the live vaccine in many affected countries. The recent refinement of the stabilate, coupled with a body of epidemiological knowledge accumulated, has helped to allay the fears over the role of the live vaccine with regards to the introduction of new *T. parva* strains [14–16]. As a result, there has been a shift in governments' policy regarding the use of live ECF vaccine in most of the affected countries. Immunization against ECF in Burundi began in 1981 with the use of a stabilate that was a combination of three local *T. parva* isolates namely; Gatumba, Gitega and Ngozi [12]. The Ngozi isolate was later removed due to its apparent association with ophthalmic problems [12]. Although attempts to test the ability of the MC vaccine to protect cross-bred calves against disease when challenged with the Burundi *T. parva* stocks were made in the early1990s, the obtained results were not conclusive [12].

Previous studies have distinguished buffalo-derived from cattle-derived *T. parva* stocks via the analysis of the p67 gene [17, 18] and shown that buffalo-derived *T. parva* have a higher diversity than the cattle-derived [14, 19, 20]. In addition, several studies have described *T. parva* antigens such as the Cytotoxic T Lymphocyte (CTL) antigens from immunized cattle [21–24]. Previous studies on the immune responses in cattle immunized with ITM have shown that MHC-I restricted CD8+ T cells are important mediators of immunity [21, 25]. In addition, studies have identified several parasite antigens recognized by CD8+ T cells as potential vaccine candidates [22, 23]. The nature and extent of polymorphism in some of these

antigens (Tp1 and Tp2) that contain defined CD8[+] T-cell epitopes have been compared in both cattle derived and buffalo derived *T. parva* isolates [18]. Furthermore, several studies on the population structure of *T. parva* in some regions of different countries using Micro- and mini-satellite markers have also revealed a high genetic diversity [13–15, 26, 27]. For example, a study on the population structure of *T. parva* in Uganda reported a mixture of genotypes in many isolates and linkage disequilibrium (LD) in three populations isolated from different areas [28]. Another study in Zambia reported a low level of genotype exchange between two districts that were geographically apart but a high level of genetic diversity within each population [29]. Further, high genetic diversity was also detected in *T. parva* isolates on a single farm in Uganda [30].

We conducted a cross sectional study with the aim of ascertaining the sero-prevalence of *T. parva* in cattle from five provinces of Burundi namely: Bujumbura, Cibitoke, Gitega, Kirundo and Cankuzo representing the Western, Central, Northern and North Eastern regions of Burundi, respectively. Furthermore, we also estimated the prevalence and determined the diversity of the p67 gene alleles as well as the extent of polymorphism in the Tp1 and Tp2 antigens of *T. parva* in the Northern, Eastern, Western, Southern and Central regions using the p104 gene PCR and sequence analysis, respectively. We further compared the antigens in field samples with those in the MC vaccine and other known *T. parva* vaccine sequences. In addition, we used a panel of 4 polymorphic minisatellites (MS7, MS19, MS29, MS35) spanning the four *T. parva* chromosomes and one microsatellite marker (ms9 on chromosome 3) to determine the population genetic analysis and sub-structuring of *T. parva* in the different regions of Burundi.

## Materials and methods

### Ethical statement

Ethical clearance for the recombinant DNA experiments in this study was approved by the University of Zambia Biomedical Research Ethics Committee (UNZABREC) under REF. 233–2019. Clearance for collection of samples was obtained from the Directorate of Animal Health, Burundi and verbal consent was sought from farmers. Blood sampling was done by a veterinarian from cattle owned by consenting farmers and did not involve endangered or protected animal species. The animals were handled humanely during sample collection.

### Study sites

Burundi is a country in east-central Africa, and is south of the Equator. The largest administrative division is the province. There are 18 provinces in Burundi, each named after its provincial capital. Samples for parasitology and serology analysis were collected between June and September 2014 from five provinces namely; Bujumbura, Cibitoke, Gitega, Kirundo and Cankuzo representing the Western, Central, Northern and North Eastern regions, respectively. Whereas for molecular and population genetic analysis, samples were collected between May to August 2017 from 11 provinces representing the five regions of Burundi. The 11 provinces were; Bubanza and Bujumbura rural from the Western region, Kirundo from the Northern region, Muramvya, Mwaro, Gitega and Kayanza from the Central region, Bururi and Makamba from the Southern region, Rutana from the Eastern region and Cankuzo from the North Eastern region (Fig 1).

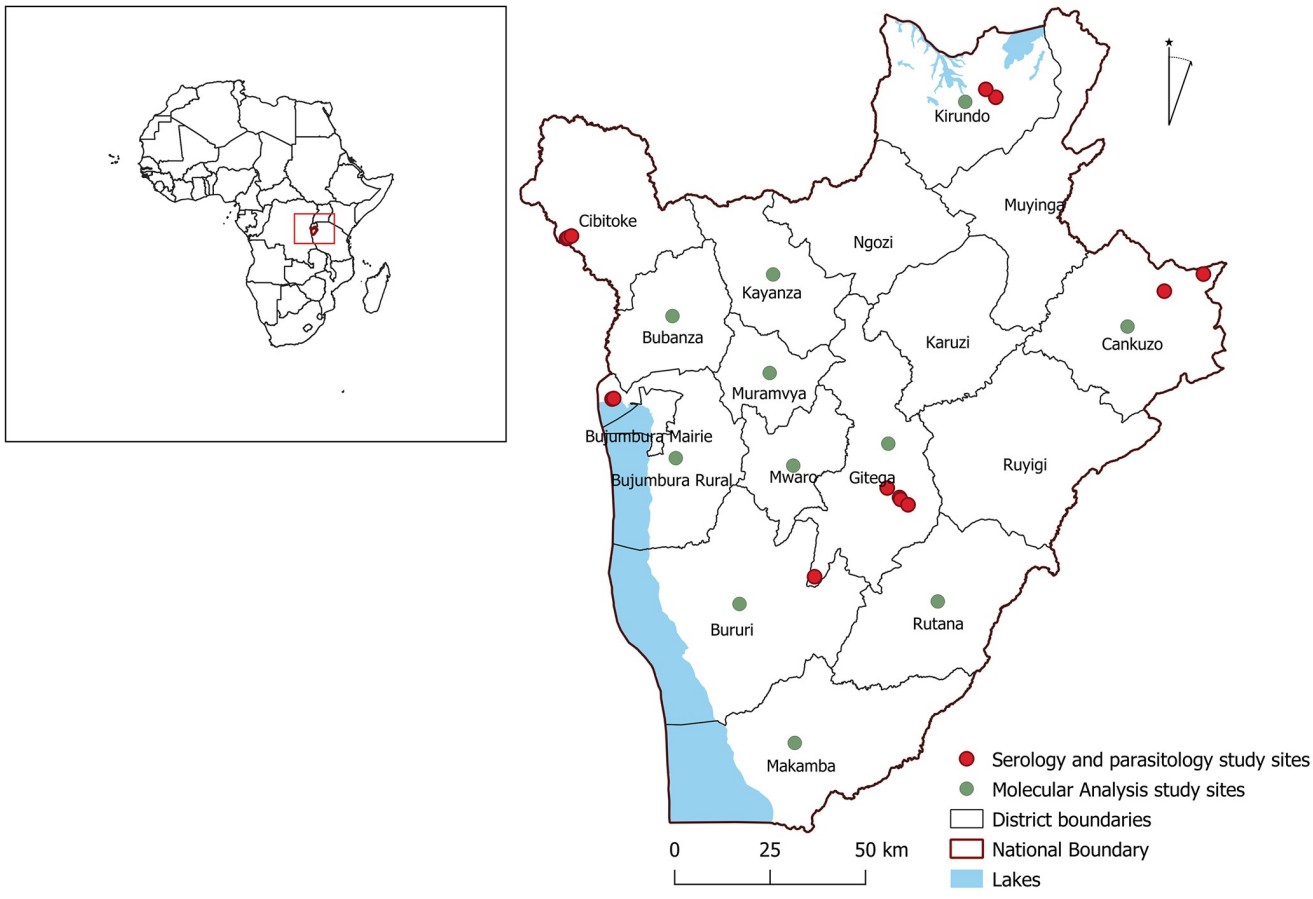

**Fig 1. Map of Burundi showing the provinces and sites where sampling of cattle was conducted.** Colour codes indicate the areas were samples for microscopic examination, serology and molecular analysis were collected. The regions were broken down as follows; Northern region (Kirundo province), Southern region (Makamba province), Eastern region (Rutana province), Western region (Bujumbura, Bubanza, Cibitoke and Bururi provinces), Central region (Muramvya, Mwaro, Gitega and Kayanza provinces) and North-Eastern region (Cankuzo province).

### Sample collection and handling

Whole blood samples were collected from the jugular vein in both EDTA and plain tubes from cattle from selected farms in the different regions of the country. The blood sample from each animal was also spotted on FTA filter papers in duplicate for future DNA extraction. Blood in vacutainers was kept on ice packs for transportation to the laboratory. Serum was collected from blood samples after centrifugation and stored at -20˚C until use. Aliquots of the sera and blood in EDTA together with spotted FTA papers were shipped to the Centre for Ticks and Tick-borne Diseases (CTTBD) in Lilongwe, Malawi, for ELISA, DNA extraction and PCR. Aliquots of the DNA extracted at CTTBD were then transported to the School of Veterinary Medicine at the University of Zambia for further PCR, sequencing and microsatellite analyses.

### *Theileria* species screening by microscopy

Thin smears (n = 1109) were made in the field and carried to the laboratory in Bujumbura for microscopic analysis. In the laboratory, the blood smears were fixed in methanol for 5 min and stained for 30 min in Giemsa stain diluted with 5% buffer. Slides were examined for intra-erythrocytic forms of *Theileria spp*. at 100X objective magnification. The smears were recorded as negative for piroplasms if no parasites were detected in 20 oil-immersion fields.

### *Theileria parva* screening by ELISA

The serum samples (n = 1,431) were screened for *T. parva* using PIM ELISA according to the protocol previously described [31].

### *Theileria parva* screening by PCR

Polymerase chain reaction (PCR) using the p104 gene primers previously described [32] was performed to detect the presence of *T. parva* genomic DNA in field samples (n = 200). The PCR mix and reaction conditions used were as described previously [27].

### PCR amplification of p67, Tp1 and Tp2 genes

*Theileria parva* p67, Tp1 and Tp2 genes were amplified from p104 gene PCR positive field samples. Primers previously described [17] were used to amplify the p67 gene using the Amplitaq Gold master mix PCR kit according to the manufacturer's instructions via a nested PCR approach. The conditions used for the initial PCR included; initial denaturation at 95°C for 10 minutes, followed by 35 cycles of denaturation at 96°C for 30 seconds, annealing at 55°C for 30 seconds and extension at 68°C for 60 seconds with a final extension step at 72°C for 5 minutes. Similar PCR conditions were used for nested PCR. Tp1 and Tp2 genes were amplified using primers previously described [18] and the PCR protocol described previously [27]. Amplicons were visualized using 1.5% agarose gel pre-stained with ethidium bromide.

### Cycle sequencing

PCR products were purified using Monofas purification kit (GL Sciences, Japan) and then subjected to Cycle sequencing using the Big Dye Terminator v3.1 sequencing kit (Life Technologies®, Applied Biosystems) according to the manufacturer's instructions. Afterwards, excess buffers and dNTPs were removed from the sequence PCR products using the ethanol precipitation method. The final purified products were then denatured and subjected to capillary electrophoresis on the ABI 3500 genetic analyzer (Applied Biosystems).

### Microsatellite PCR

The markers used in genotyping field samples are listed in S1 Table. The forward primers were labelled with fluorescent dyes and the annealing temperatures used were as described previously [33]. The PCR mix and reaction conditions were as previously described [27]. Amplified PCR products were visualized on 1.5% agarose gel coated with ethidium bromide. All the successfully amplified PCR products were then denatured and electrophoresed using the ABI Seqstudio genetic analyzer (Life Technologies). The DNA fragment sizes from the field samples, Muguga, Kiambu, Serengeti, MCL01 and Chitongo vaccine isolates were determined using the GeneMapper software ver. 5 (Applied Biosystem, Waltham, Massachusetts, USA). GeneMapper software ver. 5 scored peaks with the highest area as the most dominant allele and then using this data, a Multi-locus genotype (MLG) was constructed to represent the most dominant genotype within each sample.

### Data analysis

**ELISA analysis.**   The OD values were expressed as percent positivity (PP) relative to a reference strong-positive control serum [34]. Any test serum with a PP value of 20 or above was considered positive [31].

**Sequence analysis.**   Blast analysis using the NCBI website was employed to verify the nucleotide sequences obtained in this study. Afterwards, the sequences were assembled and

edited using genetyx ver. 12. The assembled and edited sequences were then aligned with the Muguga, Kiambu 5, Serengeti transformed, and Chitongo vaccine strain reference sequences as well as with other reference sequences from east Africa using clustal W1.6. A fasta file of this alignment was generated and converted to a MEGA file format for the purpose of constructing phylogenetic trees using MEGA ver. 6 [35]. Phylogenetic trees were constructed using 1000 bootstrap replicates as a confidence interval and for sequences that shared 100% homology, a representative sequence was used in the construction of phylogenetic trees (S2 Table). Multiple sequence alignments of amino acid sequences were also translated from the multiple sequence alignment of the nucleotides for the purpose of analyzing the CTL epitopes on both Tp1 and Tp2.

The DNA polymorphisms for each gene were calculated using DnaSP ver. 5 [36]. The single likelihood ancestor counting method using the F81 model as well as a confidence level of 0.05 on the data monkey (Available online: http://www.datamonkey.org) website was used to calculate the mean ratio of the non-synonymous substitutions vs. synonymous substitution (dN/dS) per site [37]. The distribution of genetic variation and the level of population differentiation among the sequences was investigated through the analysis of molecular variance (AMOVA) performed using GenAIEx6 [38]. Network ver. 10 was utilized to assess the haplotype similarities between the Tp1 and Tp2 nucleotide sequences of the vaccine stocks and the field samples (http://fluxus-engineering.com/). All the nucleotide sequences obtained in the current study have been deposited in DNA Data Bank of Japan (DDBJ) with designated accession numbers LC593820 to LC594067 for Tp1 and 2 and LC594555to LC 594610 for p67 (S3 Table).

**Microsatellite analysis.** Initially, the level of similarity within the MLG was analyzed using microsatellite tool kit (http://animalgenomics.ucd.i.e./sdepark/ms-toolkit/). These similarities were then visualized using allele frequency distribution and Principal component analysis (PCA) constructed using GenAIEx6 [38]. The extent of population sub-structuring was also calculated using the FSTAT computer package version 2.9.3.2 (https://www2.unil.ch/popgen/softwares/fstat.htm). LIAN (http://adenine.biz.fh-weihenstephan.de/lian) was further used to determine the null hypothesis of panmixia and linkage equilibrium. This was done by calculating the standardized index of association, the variance of pairwise differences ($V_D$), the variance of differences required for panmixia ($V_E$) and L which is the 95% confidence interval for $V_D$ [39]. Values of the index of association which are negative or closer to zero indicate panmixia (random mating) while values that are positive and significantly greater than zero indicate non panmixia (nonrandom mating). When $V_D$ value is greater than the L value, LD is indicated and the null hypothesis of panmixia is rejected and when the calculated $V_D$ is less than the L value, LE is indicated and the null hypothesis of panmixia is accepted.

## Results

### Microscopy, ELISA and PCR screening

A total of 1109 blood smears, 1431 serum and 200 DNA samples from all regions of Burundi were analysed by microscopy, ELISA and p104 gene PCR, respectively. From the blood smears, intra-erythrocytic forms of *Theileria* piroplasms were observed in 30% (332/1109) of the samples and the degree of parasitosis per region is summarised in Table 1. The ELISA results for each region are also summarised in Table 2. Overall, antibodies for *T. parva* were present in 60.1% (860/1431) of the serum samples. Furthermore, *T. parva* genomic DNA was detected in 79% (158/200) of the DNA samples selected from the samples originating from the Western, Northern, Central, North-eastern, Eastern and Southern provinces using the *T. parva* specific p104 gene PCR (Table 3).

**Table 1. Proportion of *Theileria* species-like blood smear positive samples collected from the different sites in Burundi.**

| Region | Province | Number of blood smears | Positive blood smears | Percent (%) |
|---|---|---|---|---|
| **Western** | Cibitoke | 200 | 78 | 39 |
| | Bujumbura rural | 200 | 86 | 43 |
| **Central** | Gitega | 343 | 47 | 13.7 |
| **Northern** | Kirundo | 200 | 78 | 39 |
| **North-Eastern** | Cankuzo | 166 | 43 | 26 |
| **Total** | | 1109 | 332 | 30 |

**Table 2. Proportion of *Theileria parva* antibody positive samples collected in different provinces in Burundi.**

| Region | Province | Number of serum samples | ELISA positive | Percentage (%) |
|---|---|---|---|---|
| **Western** | Cibitoke | 199 | 101 | 50.8 |
| | Bujumbura rural | 270 | 166 | 61.5 |
| **Central** | Gitega | 470 | 285 | 60.6 |
| **Northern** | Kirundo | 161 | 100 | 62.1 |
| **North-Eastern** | Cankuzo | 331 | 208 | 62.8 |
| **Total** | | 1431 | 860 | 60.1 |

## Sequence and phylogenetic analysis of the p67 gene

The 2946 to 3767 bp region of the p67 *T. parva* gene (Accession number M67476) consisting of an exon-intron-exon was sequenced from 64/158 p104 gene PCR positive field samples. Currently, four alleles of p67 are recognised [40] and designated as allele 1 with a 129 bp deletion and allele 2 without the deletion [17], Allele 3 with the 179 bp deletion and Allele 4, similar in sequence to allele 3 but without the deletion [40]. Phylogenetic analysis showed four clusters namely A, B, C and D (Fig 2). Within these clusters (Fig 2), sequences clustered according to the p67 allele types as previously described [40]. All the field sample sequences formed a cluster with Muguga M67476 allele 1 and KY912962 allele 1 reference sequences in cluster A. None of the field samples clustered with any of the remaining alleles thus within the samples analysed, only allele type 1 which is present in Muguga was represented.

**Table 3. Proportion of *Theileria parva* p104 PCR positive samples collected from different provinces in Burundi.**

| Region | Province | Number of DNA samples | p104 PCR positive | Percentage (%) |
|---|---|---|---|---|
| **Western** | Bubanza | 25 | 22 | 88 |
| | Bujumbura | 52 | 49 | 94.2 |
| **Central** | Gitega | 5 | 4 | 80 |
| | Kayanza | 16 | 13 | 81.3 |
| | Muramvya | 8 | 8 | 100 |
| | Mwaro | 10 | 10 | 100 |
| **Northern** | Kirundo | 32 | 23 | 71.9 |
| **North-Eastern** | Cankuzo | 4 | 2 | 62.8 |
| **Southern** | Bururi | 8 | 2 | 50 |
| | Makamba | 5 | 2 | 40 |
| **Eastern** | Rutana | 35 | 23 | 65.7 |
| **Total** | | 200 | 158 | 79.0 |

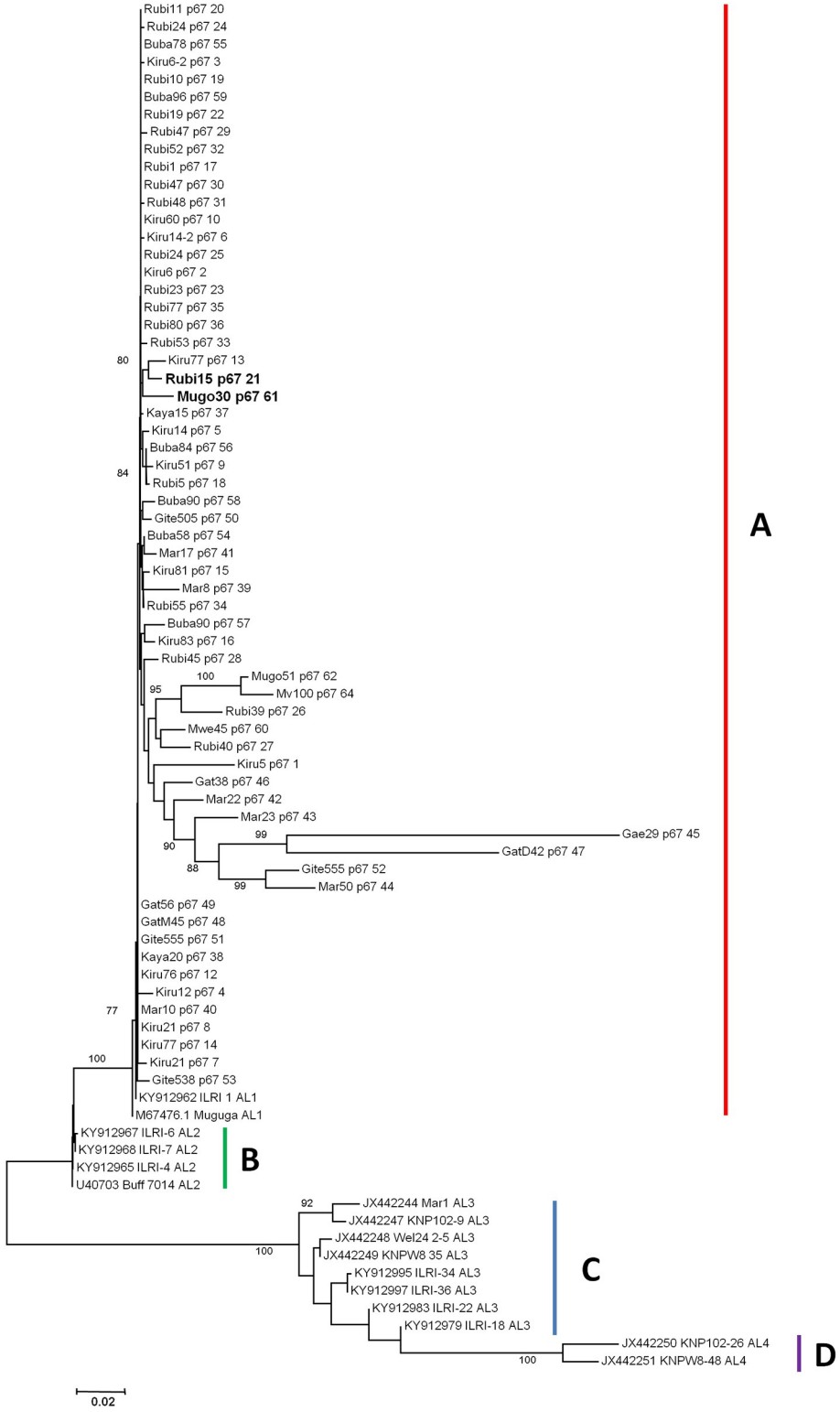

**Fig 2. p67 neighbour joining phylogenetic tree based on 821 bp nucleotide sequences from Burundi as well as reference sequences from east and Southern Africa.** The phylogenetic tree was constructed using MEGA ver. 6 with 1000 bootstrap replicates as a level of confidence.

## Sequence diversity and phylogenetic analysis of the Tp1 gene locus

The complete protein encoding Tp1 CTL gene (432 bp) of *T. parva* was sequenced from 130 samples out of the 158 p104 gene positive samples. The epitopes from the majority of the field samples (115/130) had 100% identity with the MC epitope (VGYPKVKEEML) reported previously [18]. Further, 7 samples possessed the epitope VGYPKVKEEII present in Chitongo and previously reported in Kenya [18] while 7 other samples possessed the epitope VGYPKV-KEEM**I** also previously reported in Kenya [18] and in Rwanda [27]. Only a single sample (Kayanza 17) possessed an epitope (VGYPKVKEEM**V**) that was different from any of the known vaccine stocks and has not been reported anywhere else previously. Overall, MC epitopes accounted for 88.46% of the epitopes identified and thus was the most abundant in the field samples from Burundi. The field sequences had a DNA polymorphism of 0.73% and the mean ratio of the dN/dS of 0.674 with two purifying selection sites and no positive selection sites.

Phylogenetic analysis of Tp1 gene showed five (5) major clusters (Fig 3). Clusters A, B, C and D consisted of cattle derived sequences while cluster E comprised of Buffalo derived sequences (Fig 3). Cluster A contained the majority of the samples (n = 87) and was further divided into minor clusters A-I, A-II, A-III and A-IV. Within minor cluster A-I, twenty two samples clustered with the Muguga, Kiambu, Serengeti and Nyakizu vaccine isolates as well as reference sequences from Tanzania, Kenya and South Sudan (Fig 3). In minor cluster A-IV, six samples clustered with a single reference sequence from South Sudan. The remaining minor clusters A-II (n = 54) and A-III (n = 5) exclusively consisted of samples from the different regions of Burundi (Fig 3). Cluster B was also divided into two minor clusters namely B-I, exclusively comprising field samples (n = 13) and B-II, comprising field samples (n = 6) clustering with Chitongo and reference samples from Tanzania (Fig 3). Cluster D comprised field samples (n = 3) and reference sequences from South Sudan and Tanzania as well as the Gikongoro vaccine isolate while cluster C consisted of field samples (n = 20) and a reference sequence from Kenya (Fig 3). Two field samples did not share any clusters with either other field samples or reference sequences. None of the field samples clustered with buffalo-derived sequences in cluster E and there was no evidence of clustering of field samples according to the origin of the sample (Fig 3).

Genetic analysis of Tp1 using the analysis of molecular variance (AMOVA) revealed that most of the genetic variation observed was within the population (94%) while the least variation was among the populations (6%). A total of 62 haplotypes were identified for Tp1 gene on median-joining network analysis (Fig 4). Haplotype 9 (H9) had the highest frequency and comprised of 24 samples originating from the Western (n = 11), Northern (n = 7), Central (n = 4) and Eastern (n = 2) regions of Burundi. This was followed by H1 (n = 18) comprising Muguga, Kiambu, Serengeti and field samples from the Western (n = 6), Central (n = 5), Northern (n = 3) and Eastern (n = 1) regions and H10 (n = 16), comprising samples from the Northern (n = 1), Central (n = 9), Western (n = 4), Southern (n = 1) and Eastern (n = 1) regions. Haplotype H6 and H25 each comprised 5 field samples while H33 comprised of four samples. The field samples in haplotypes H6, H25 and H33 originated from the Western (n = 3) and Eastern (n = 2) regions, the Western (n = 1), Central (n = 2) and Northern (n = 2) and the Eastern (n = 2), Western (n = 1) and Northern (n = 1) regions, respectively. The remaining haplotypes comprised either 1, 2 or 3 field samples. Haplotype H1, comprising MC, was indirectly linked to H10 through H4 (Nyakizu), H5 and H7. Haplotype H10 was both directly and indirectly linked to H9 via H6, H28 and H55 and also through a pair of median vectors. With the exception of Muguga, Kiambu and Serengeti vaccine isolates, none of the other vaccine isolates had similar haplotypes with the field samples. Furthermore, similar haplotypes were also present in

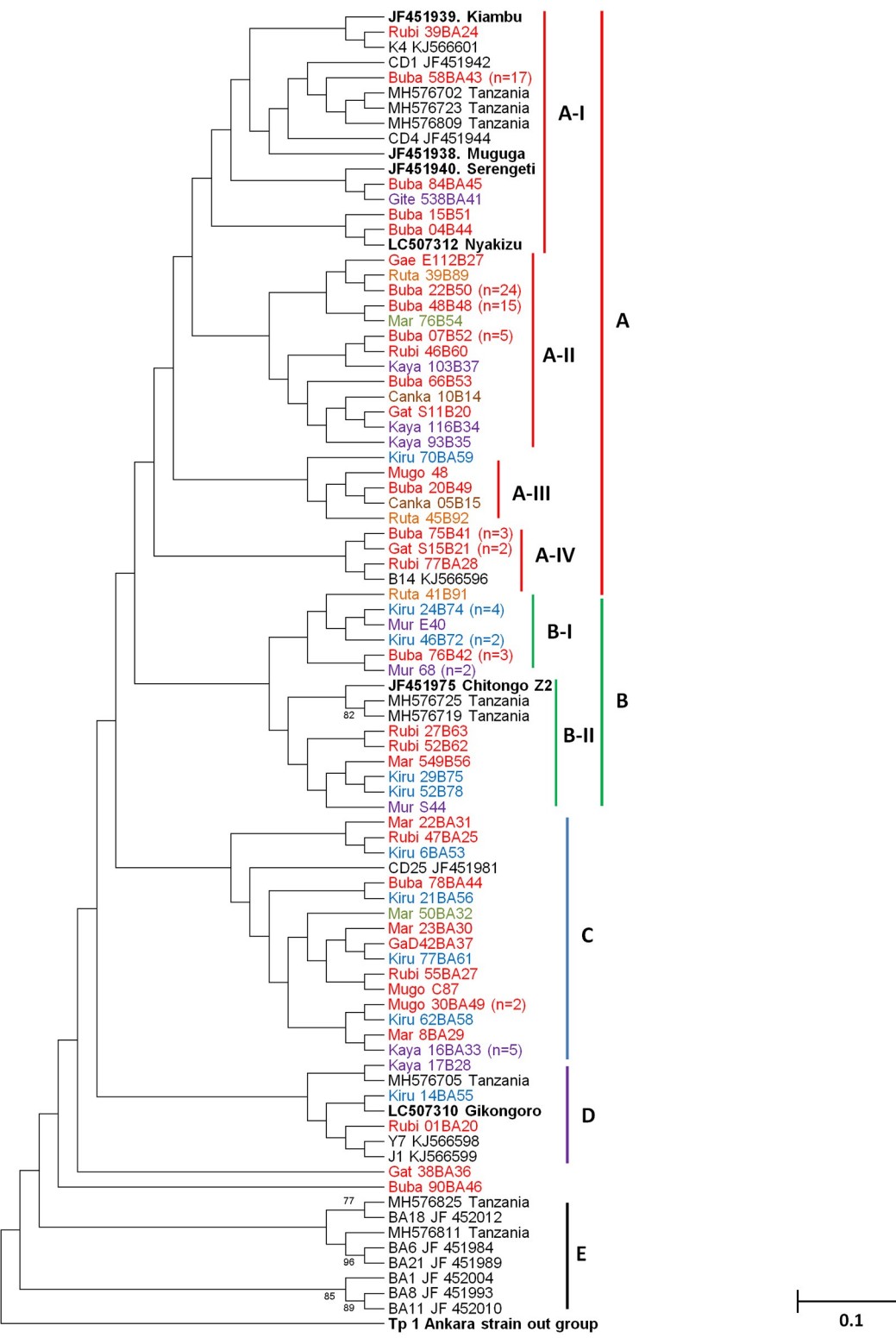

**Fig 3. Neighbour joining phylogenetic tree for *Theileria parva* CTL Tp1.** The tree was generated from 432 bp nucleotide sequences using MEGA ver. 6 with 1000 bootstrap replicates as a confidence level. For sequences that shared 100% identity, a single sequence was used to represent the group on the phylogenetic tree. Samples from the Northern, Southern, Eastern, Western, Central and North Eastern regions are given in blue, black, orange, yellow, purple and brown colour codes, respectively.

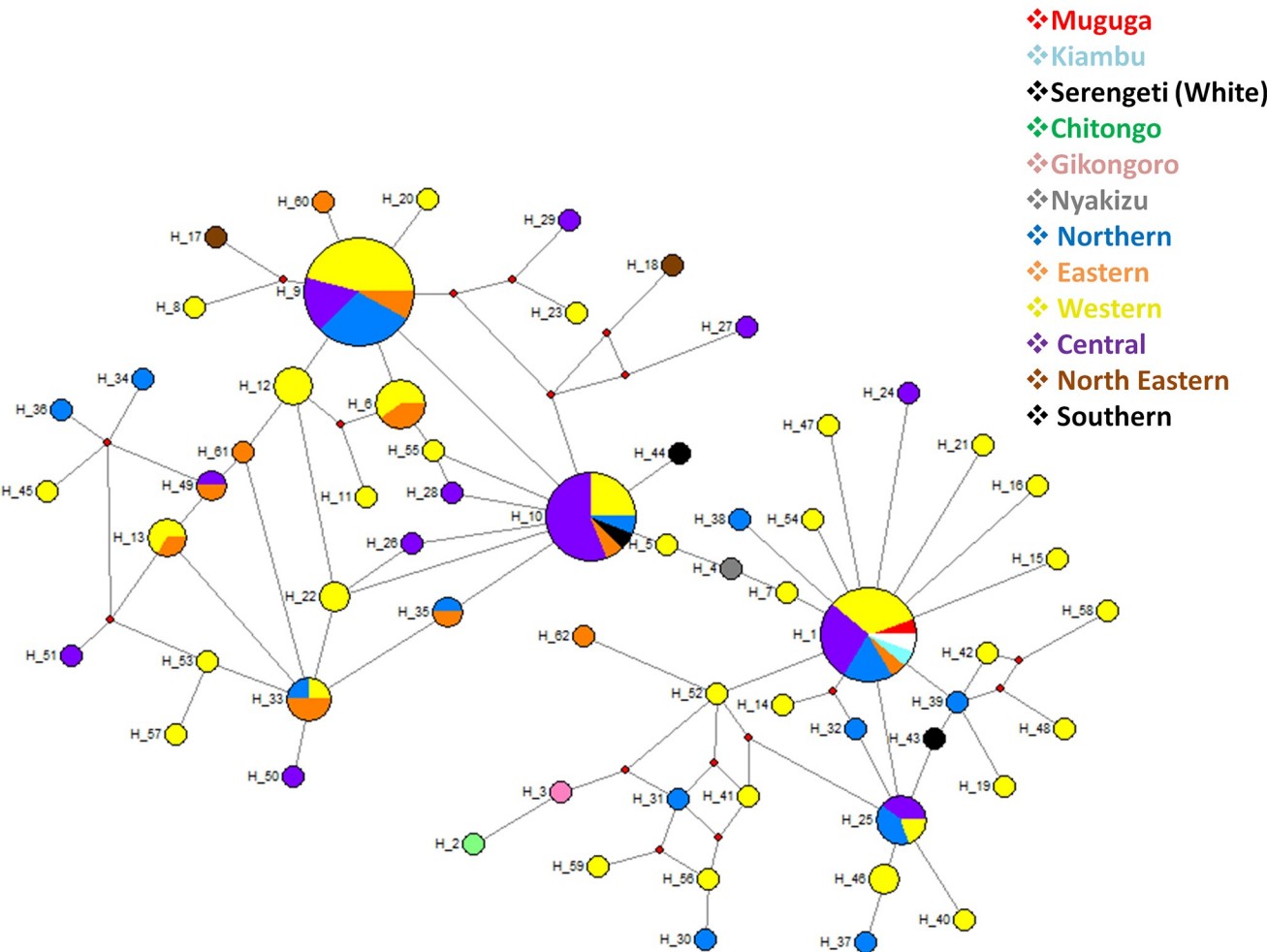

**Fig 4. Median-joining network of the *Theileria parva* Tp1 gene constructed using network 5 and based on the polymorphic sites of Tp1.** The median joining network shows a star-like radiation pattern and the sizes of the circles correspond to the haplotype frequency. The color codes denote the origin of the samples.

field samples from different geographical regions and the star-like appearance on the abundant haplotypes indicated evidence of population expansion (Fig 4).

## Sequence diversity, phylogenetic and similarity analyses of Tp2 gene locus

The Tp2 gene (531 bp) of *T. parva* was successfully sequenced from 118 samples out of the 158 p104 PCR positive samples. This region encodes the 174 amino acid protein of the Muguga reference sequence (XP_765583) and comprises six epitopes designated as CTL 1 (SHEELKKLGML), CTL 2 (DGFDRDALF), CTL 3 (KSSHGMGKVGK), CTL 4 (FAQSLVCVL), CTL 5 (QSLVCVLMK) and CTL 6 (KTSIPNPCKW) [22, 24]. Out of 118 sequences, a total of 58, 66, 67, 73, 70 and 87 samples had 100% identity with Muguga, Kiambu and Serengeti vaccine stocks (making up the Muguga cocktail vaccine) on epitopes 1, 2, 3, 4, 5 and 6, respectively. The total number of epitopes that were different from those in Muguga, Kiambu and Serengeti on epitopes 1, 2, 3, 4, 5 and 6 were 14, 20, 13, 11, 20 and 16, respectively (Table 4). In addition, a total of 38, 45, 47, 43, 47 and 35 samples possessed epitopes that were different from Muguga, Kiambu and Serengeti on epitopes 1, 2, 3, 4, 5 and 6, respectively (Table 4). On the other hand, the total number of samples that possessed 100% identity with

**Table 4. Tp2 epitopes identified in this study from unvaccinated cattle from the different regions of Burundi that differ from Muguga cocktail epitopes.**

| Tp 2 Gene | CTL 1 | CTL 2 | CTL 3 | CTL 4 | CTL 5 | CTL 6 |
|---|---|---|---|---|---|---|
| | SEEELNKLGML (1) | DDFDRNALF (1) | KSSHGMGKIGK (1) | FAQSIMCVL (1) | QSLVCVLRK (1) | KPSIPNPCKW (1) |
| | SHNELKNLGML (1) | DGSDKNTLF (1) | LSSHGMGKVGK (6) | FAPSIKCVL (2) | QSLVCVLLK (1) | KTSNPNPCKW (1) |
| | IDEELNKLGML (1) | EGFDKEKLF (3) | ISSHGMGKVGK (1) | FAQSLMCVL (4) | QSIMCVLLK (1) | KPNVPNPCRW (1) |
| | SDEELNKLGMV (1) | NDLDRNTLF (1) | KSPHGMGKVGK (1) | FVQSIMCVI (2) | PSIKCVLHQ (1) | ITDIPNPCKW (1) |
| | SDEELENLGLL (1) | DDFDRDALF (1) | KSSHGMGKIGR (1) | FAQSIMCVV (1) | QSLMCVLMK (2) | ETDIPNPCKW (1) |
| | TEEELKKMGMV (3) | DGLDRNALF (1) | LSSHGMGKVGK (1) | FAQSLMCVS (1) | QSIMCVINK (2) | KPSVPNPCKW (1) |
| | SDDELDTLGLL (1) | EDFDRNTLF (1) | KSSTCMGIVGR (1) | FAPSIKCVS (2) | QSLMCVSQK (1) | KTSVPNPCEW (1) |
| | SDDELDNLGML (3) | DGFDRDTLF (2) | KSSKSMGIVGR (2) | LAPSIKCVS (2) | QSLVCVLLQ (1) | KPSVPNPCKW (2) |
| | SDDELDNLGMV (1) | QDFDRNTLF (1) | LPSHGMGKIGK (1) | LAASIRCVS (1) | PSIKCVLPQ (1) | KTRFPNPCKW (1) |
| | SDNELDTLGLL (20) | EGFDKDTLF (1) | LTSHGMGKIGR (22) | LAASIKCVS (23) | QSLMCVLQK (1) | VNDIPNPCKW (2) |
| | SDDELNNLGML (2) | PDLDKNTLF (2) | LSSHGMGKIGR (7) | FAASIKCVS (4) | QSLMCVINK (1) | KTDIPNPCKW (1) |
| | SDDELEKLGLL (1) | PDPDKNTLF (1) | LSSHGMGRIGK (1) | | QSIMCVVLK (1) | KTSIPNPCEW (1) |
| | SDDELKKLGLL (1) | PDFDRNAL F (1) | LSSHGMGRIGR (2) | | ASIRCVSHH (1) | KPSVPIHCKW (1) |
| | SDEELNNLGLV (1) | HDLDKNRLF (19) | | | PSIKCVSHQ (1) | KPSVPNPCEW (18) |
| | | HDLDKNTLF (2) | | | PSIKCVSQN (1) | KPSVPNPCRW (1) |
| | | RDLDRNTLF (2) | | | PSIKCVSQH (1) | KPSVPNPCER (1) |
| | | HGLDRNTLF (1) | | | PSIKCVSHH (1) | |
| | | HGLDKNTLF (1) | | | ASIKCVSHH (24) | |
| | | PDLDRNTLF (2) | | | ASIKCVSQH (1) | |
| | | EDLDKDTLF (1) | | | ASIKCVSQY (3) | |
| **Total number of epitopes/locus** | 14 | 20 | 13 | 11 | 20 | 16 |
| **Total number of samples with different epitopes/locus** | 38 | 45 | 47 | 43 | 47 | 35 |

Epitopes identified in Burundi field samples. The number of samples possessing each respective epitope is given in brackets. Polymorphic amino acids on each epitope are presented in bold.

Chitongo vaccine isolate on epitopes 1, 2, 3, 4, 5 and 6 was only 13, 12, 10, 7, 7 and 15, respectively. Compared to Chitongo, the epitopes from Muguga, Serengeti and Kiambu, collectively referred to as the Muguga cocktail were highly represented in the field samples. Overall, the Muguga, Kiambu and Serengeti (contained in the Muguga cocktail) epitopes were the most abundant compared to the Chitongo epitopes. In a similar manner with Tp1, the epitopes did not divide according to the region of origin of the sample. The calculated DNA polymorphism from the sequences from Burundi was 11.8% and the mean ratio of dN/dS was 0.640. A total of 8 diversifying/positive and 30 negative/purifying selection sites were also observed.

Phylogenetic analysis of Tp2 gene showed four (4) main clusters namely; C, D, E and F (Fig 5). Samples from different regions of Burundi were observed in clusters C, D and E and none in cluster F which exclusively contained Buffalo derived sequences (Fig 5). Cluster C was further divided into five (5) minor clusters namely C-I, C-II, C-III and C-IV (Fig 5). Minor cluster C-I, mainly comprised field samples (n = 8), reference sequences from South Sudan and Kenya as well as the Muguga, Serengeti and Kiambu vaccine isolates. On the other hand, cluster C-II comprised only field samples (n = 3) and reference sequences from South Sudan, Kenya and Tanzania. Cluster C-III comprised the Nyakizu isolate together with field samples (n = 39) while cluster C-IV exclusively contained field samples (n = 19). Of the remaining field samples (n = 5) in cluster C, only a single sample closely clustered with Gikongoro vaccine isolate while four were only related to Gikongoro. Cluster D also exclusively comprised field samples (n = 8) from the Western (n = 3), Eastern (n = 2) and Central (n = 3) regions of Burundi (Fig 5). Sequences in cluster E could be divided into two minor clusters E-I and E-II. The field samples (n = 11) in E-I clustered with the Chitongo vaccine isolate while those (n = 23) in cluster E-II clustered with a reference sequence from Kenya (Fig 5). The remaining four field samples did not form a cluster with the other field samples or reference sequences but were closely related to minor cluster E-I (Fig 5). Clustering of samples according to the region of origin was not observed and none of the Burundi samples clustered with any buffalo-derived strains (Fig 5). Furthermore, analysis of molecular variance (AMOVA) of the Tp2 gene showed that the genetic variation observed was mostly within the population (99%) and the least variation was among the populations (1%).

A total of 80 haplotypes were identified on Tp2 gene as presented in the median joining network (Fig 6). Haplotype H9 (n = 32) had the highest frequency followed by haplotypes H1 (n = 6), H26 (n = 6), H10 (n = 5) and H24 (n = 5). The remaining haplotypes comprised a single field or vaccine isolate sample. Muguga and Serengeti vaccine isolates were present in H1 (n = 6) together with field samples from Western (n = 2), Eastern (n = 1) and Central (n = 1) regions while Kiambu, Chitongo, Gikongoro and Nyakizu vaccine isolates were represented by haplotypes H2, H3, H4 and H5, respectively. H9 comprised field samples from the Western (n = 11), Central (n = 9), Northern (n = 6) and Eastern (n = 6) regions while H10 comprised field samples from the North-Eastern (n = 1), Central (n = 2), Eastern (n = 1) and Western (n = 1) regions. Further, H24 and H26 comprised of field samples from Western (n = 3), Northern (n = 1) and Southern (n = 1), and Central (n = 3), Eastern (n = 1) and Western (n = 2) regions, respectively. Haplotype H9 formed the anchor of the network from which several haplotypes were observed to radiate. The haplotypes H1 comprising the Muguga and Serengeti vaccine isolates and the H5 comprising of Nyakizu vaccine isolate was directly linked to H9 while other haplotypes comprising vaccine isolates were indirectly linked to H9. Particularly, H2 and H4 were indirectly linked to H9 via H26 and H10. Haplotype H3 comprising the Chitongo vaccine isolate was the most distantly related haplotype to H9 among the vaccine isolates. In addition, the radiating pattern from the dominant haplotype H9 observed on the network indicates population expansion. Furthermore, the MJ network indicates high diversity of Tp2 gene with no evidence of restriction of haplotypes to their respective geographical regions in Burundi (Fig 6).

## Marker diversity, allelic variation and similarity of Burundi field samples to Muguga cocktail and Chitongo vaccine isolates

A panel of four (MS39, MS25, MS7 and MS19) microsatellite and one mini-satellite (ms9) marker encompassing the 4 chromosomes of *T. parva* were used to genotype 65 field samples collected from the Western, Northern, Central, Eastern and Southern regions of Burundi as

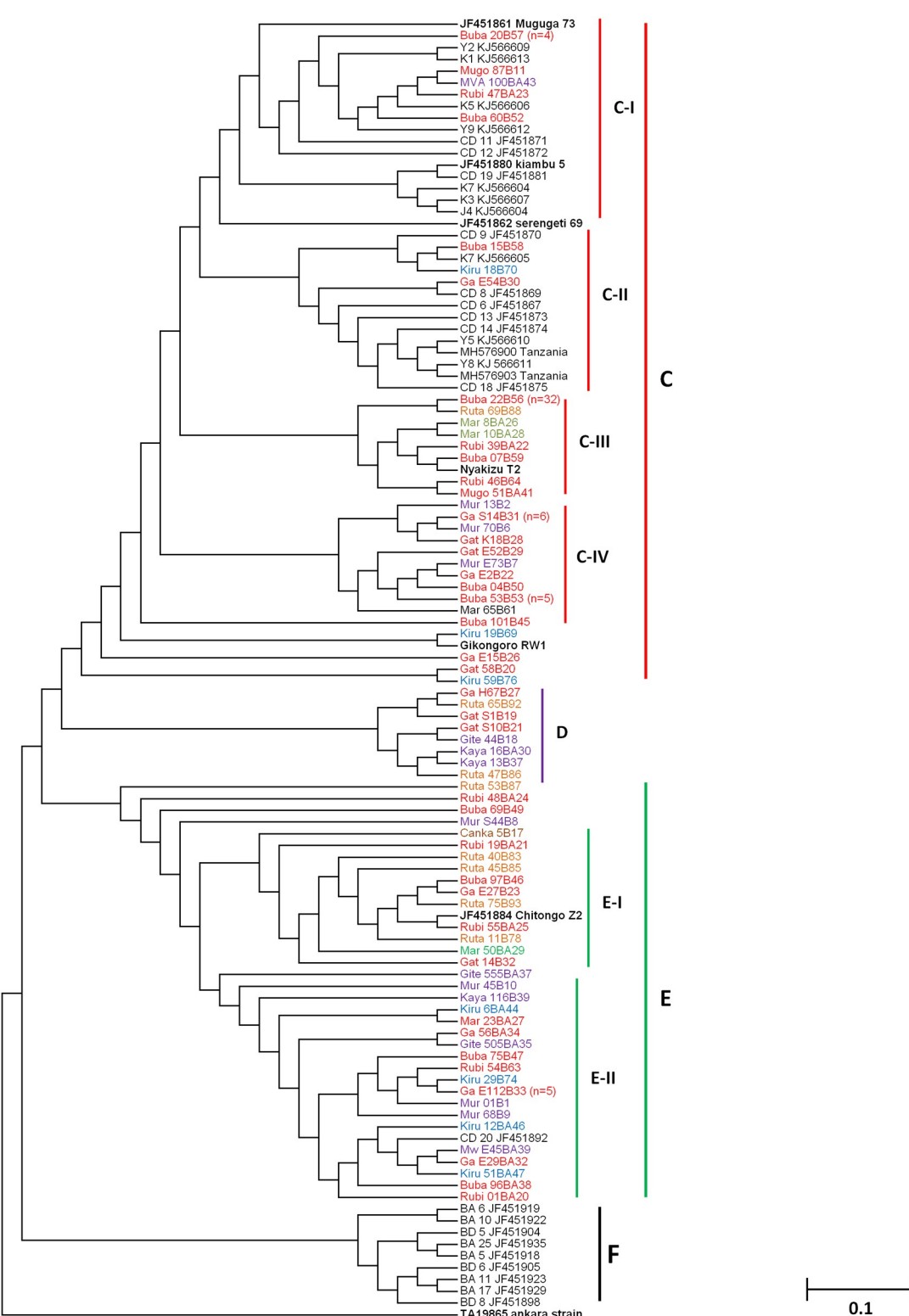

**Fig 5.** *Theileria parva* **CTL Tp2 neighbour joining phylogenetic tree generated from 531 bp nucleotide sequences.** MEGA ver. 6 with a confidence level of 1000 bootstrap replicates was used to generate the phylogenetic tree. Samples from the Northern, Southern, Eastern, Western, Central and North Eastern regions are given in blue, black, orange, yellow, purple and brown colour codes. Only a single sequence was used to represent sequences that shared 100% homology.

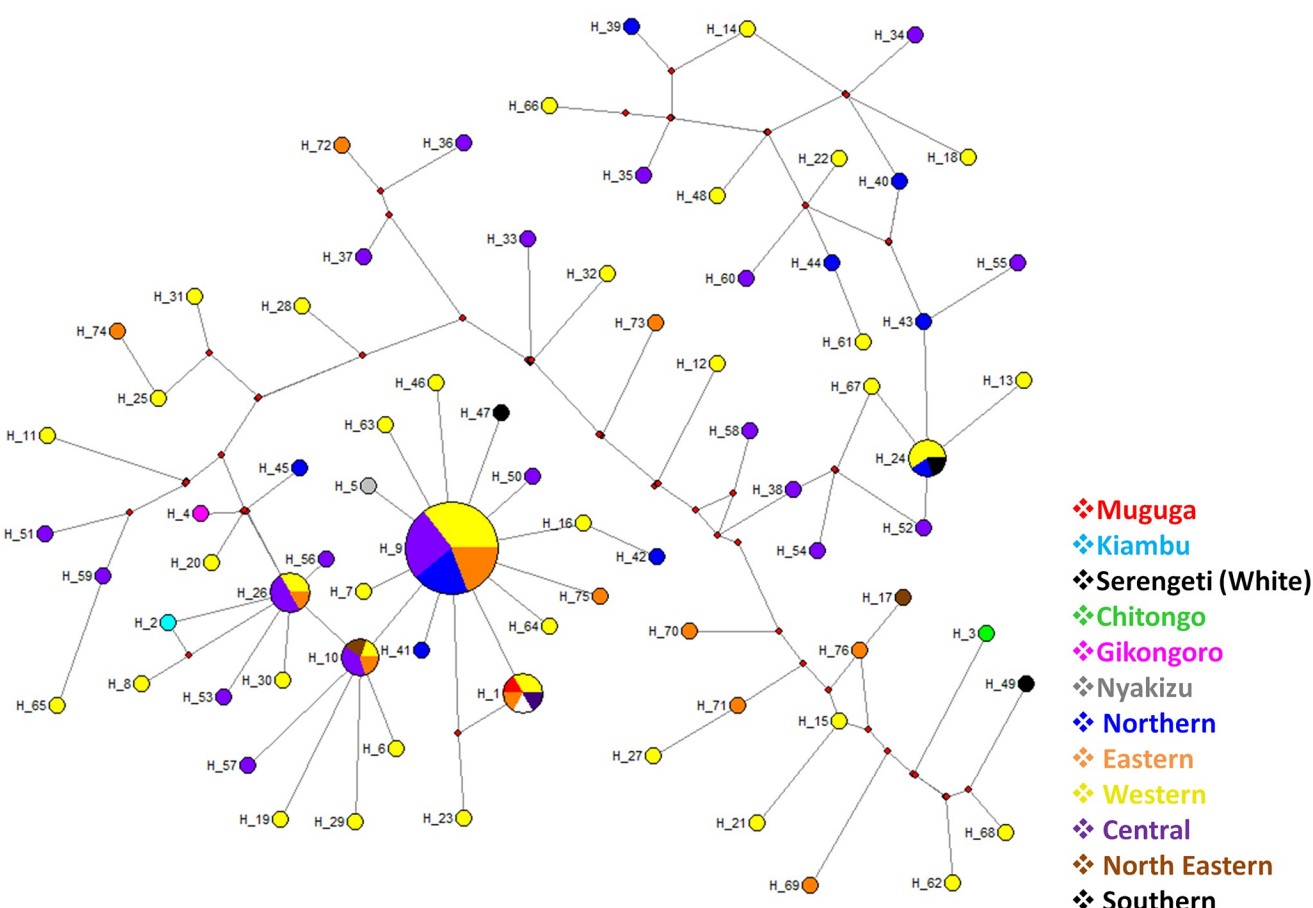

**Fig 6. Tp2 network median-joining network based on the polymorphic sites of Tp2 from Burundi showing star-like radiation pattern of haplotypes.** The sizes of the circles correspond to haplotype frequencies. The colour codes denote the places of origin of the samples.

well as the Muguga, Serengeti, Kiambu and the MCL01, collectively referred to as the Muguga cocktail (MC) population and the Chitongo vaccine isolate. Marker MS39 produced the highest number (n = 42) of alleles while marker MS19 produced the least (n = 27) thus marker MS39 was the most polymorphic (Table 5). In the MC population, 4 was the highest number of alleles while 2 was the least and with regards to Chitongo, 3 was the highest and 1 was the least. Similar and high gene diversities were observed across the different populations except for Chitongo which recorded a gene diversity of 0.00 (Table 5). The gene diversities recorded from the Western, Northern and Central regions as compared to those from Eastern and Southern regions were higher (Table 5). The reduced gene diversities recorded in the Eastern and Southern regions as compared to those observed in the Western, Northern and Central regions could have been due to the reduced sample size for the Eastern and Southern regions, and as such their results must be treated with caution. Furthermore, Muguga cocktail population also showed higher gene diversity as compared to Chitongo population.

Allele frequency distribution showed presence of both shared and unique alleles on each loci among the field samples (Fig 7, S1 and S2 Figs). Furthermore, MC population only shared a total of 1 allele with field samples on loci MS19 and ms9 while Chitongo population did not share any alleles with field samples on any loci (Fig 8, S3 and S4 Figs). Within the Burundi field sample populations, 20 alleles were shared across the different regions while 32, 16, 70, 23

**Table 5. Genetic diversity and number of alleles of *Theileria parva* from, Burundi field samples, Muguga cocktail and Chitongo vaccine stocks.**

| | Population | N | MS39 | MS7 | MS25 | MS19 | ms9 |
|---|---|---|---|---|---|---|---|
| Number of Alleles | Muguga Cocktail | 6 | 4 | 3 | 3 | 2 | 4 |
| | Chitongo | 5 | 1 | 2 | 3 | 1 | 3 |
| | Western | 25 | 18 | 19 | 17 | 14 | 16 |
| | Central | 13 | 9 | 9 | 8 | 7 | 8 |
| | Northern | 11 | 8 | 9 | 9 | 10 | 10 |
| | Eastern | 8 | 5 | 4 | 3 | 6 | 3 |
| | Southern | 8 | 7 | 5 | 6 | 4 | 5 |
| | Overall | 76 | 42 | 36 | 33 | 27 | 39 |
| Gene Diversity | Muguga Cocktail | 6 | 0.867 | 0.733 | 0.733 | 0.333 | 0.867 |
| | Chitongo | 5 | 0.00 | 0.400 | 0.800 | 0.00 | 0.800 |
| | Western | 25 | 0.970 | 0.977 | 0.967 | 0.940 | 0.953 |
| | Central | 13 | 0.936 | 0.949 | 0.923 | 0.910 | 0.936 |
| | Northern | 11 | 0.945 | 0.964 | 0.964 | 0.982 | 0.982 |
| | Eastern | 8 | 0.893 | 0.821 | 0.750 | 0.893 | 0.750 |
| | Southern | 8 | 0.964 | 0.893 | 0.929 | 0.750 | 0.857 |

and 25 unique alleles were observed from Northern, Eastern, Western, Central and Southern populations. Overall, a total of 156, 13 and 9 unique alleles were also observed from field samples, MC and Chitongo populations, respectively (Fig 8). The higher proportion of unique alleles compared to shared alleles within the field samples from the different regions of Burundi and also between the field samples and MC indicates the possibility of the presence of geographical or genetic sub-structuring (Fig 8). The same could be attributed to sub-structuring between field samples and the Chitongo population.

## Similarity analysis between Muguga cocktail and field samples

The level of similarity and sub-structuring among the Burundi populations, MC and Chitongo was further assessed using PCA (Fig 9). Field samples from the Western, Northern, Central

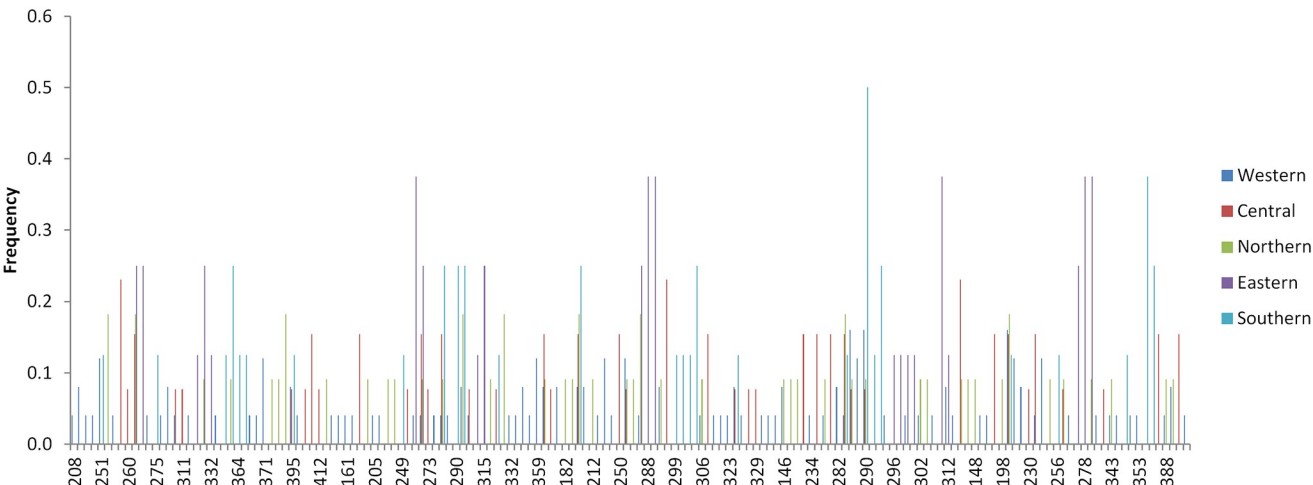

**Fig 7. Overall frequency of alleles from Burundi field samples without the Muguga cocktail and Chitongo vaccine stocks.** Both shared and unique alleles are present and the predominant alleles were calculated as proportions of the total of each satellite marker. The histograms were generated from the multi-locus genotype.

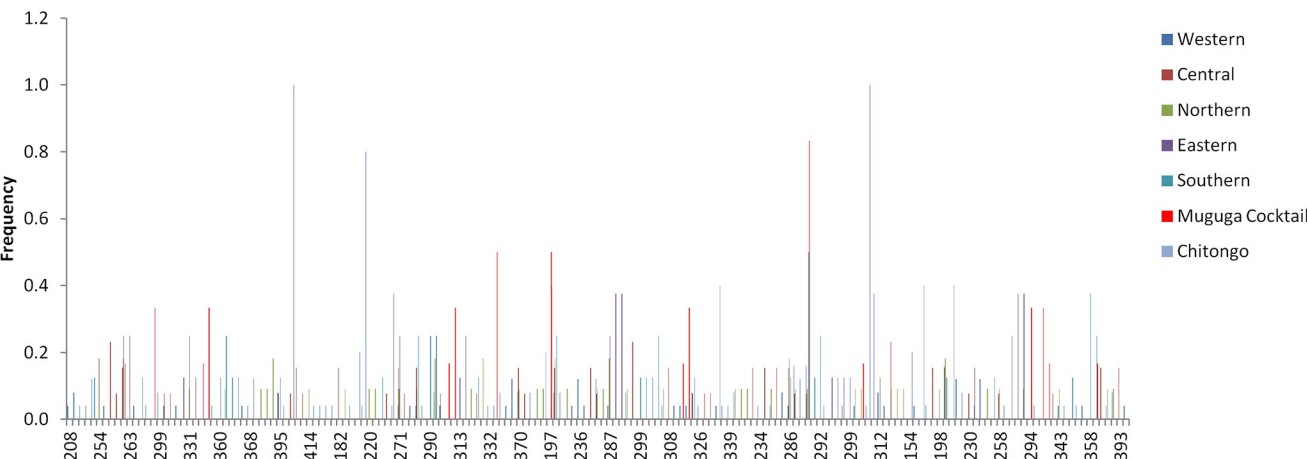

**Fig 8. Overall allele frequencies from Burundi field samples together with the Muguga cocktail and Chitongo vaccine stocks showing the presence of both shared and unique alleles.** Predominant alleles were calculated as proportions of the total of each satellite marker and histograms were generated from the multi-locus genotype.

and Southern populations were distributed in all quadrants while those from the Eastern population were more restricted to the upper and lower right quadrants (Fig 9). Adding MC in the analysis divided the field samples into two populations, a smaller (A) and a larger (B) population (Fig 10). The smaller population comprised three MC samples together with field samples from Southern, Western, Northern and Central while the larger population consisted of the majority of field samples from all populations and two MC samples (MCL01). When Chitongo was introduced in the analysis, marked separation of the Chitongo population from field samples and MC population was observed (Fig 11) indicating a substantial difference between Chitongo and the rest of the populations under study (Fig 11).

## Population differentiation, diversity and genetic analysis

The highest estimated heterozygosity and mean number of genotypes was observed in the western region while the lowest was observed in the Eastern region possibly due to the reduced sample size (Table 6). When all regions were treated as a single population, a higher estimated heterozygosity of 0.912 and mean number of genotypes of 7.434 were observed. The overall mean number of genotypes per loci and estimated heterozygosity for all the populations under study, MC inclusive, was 0.878 and 6.652, respectively (Table 6). The level of sub-structuring observed on both allele frequency distribution and PCA was further assessed using Wright's F index calculated via the Fstat computer package ver. 2.9.3.2. (https://www2.unil.ch/popgen/softwares/fstat.htm). The $F_{ST}$ value of 0.047 indicating low genetic differentiation was obtained when all populations from the Western, Northern, Central, Eastern and Southern regions were treated as a single population. When the sub-structuring between population A and B was assessed, an $F_{ST}$ value of 0.114, indicating moderate genetic differentiation was observed. In addition, combining field samples with MC produced an $F_{ST}$ value of 0.065, indicating low to moderate genetic differentiation. Furthermore, to determine the level of panmixia within the samples under study, linkage equilibrium levels of the alleles at all loci pairs were measured using the index of association. In the Western and Northern regions, the index of association was -0.0024 and 0.0303, and the $V_D$ values of 0.1832 and 0.1764 which were less than the L values of 0.2100 and 0.2135 were obtained, respectively, implying the presence of linkage equilibrium and panmixia (random mating) (Table 6). For the remaining regions namely; Central,

**Principal Coordinates (PCoA)**

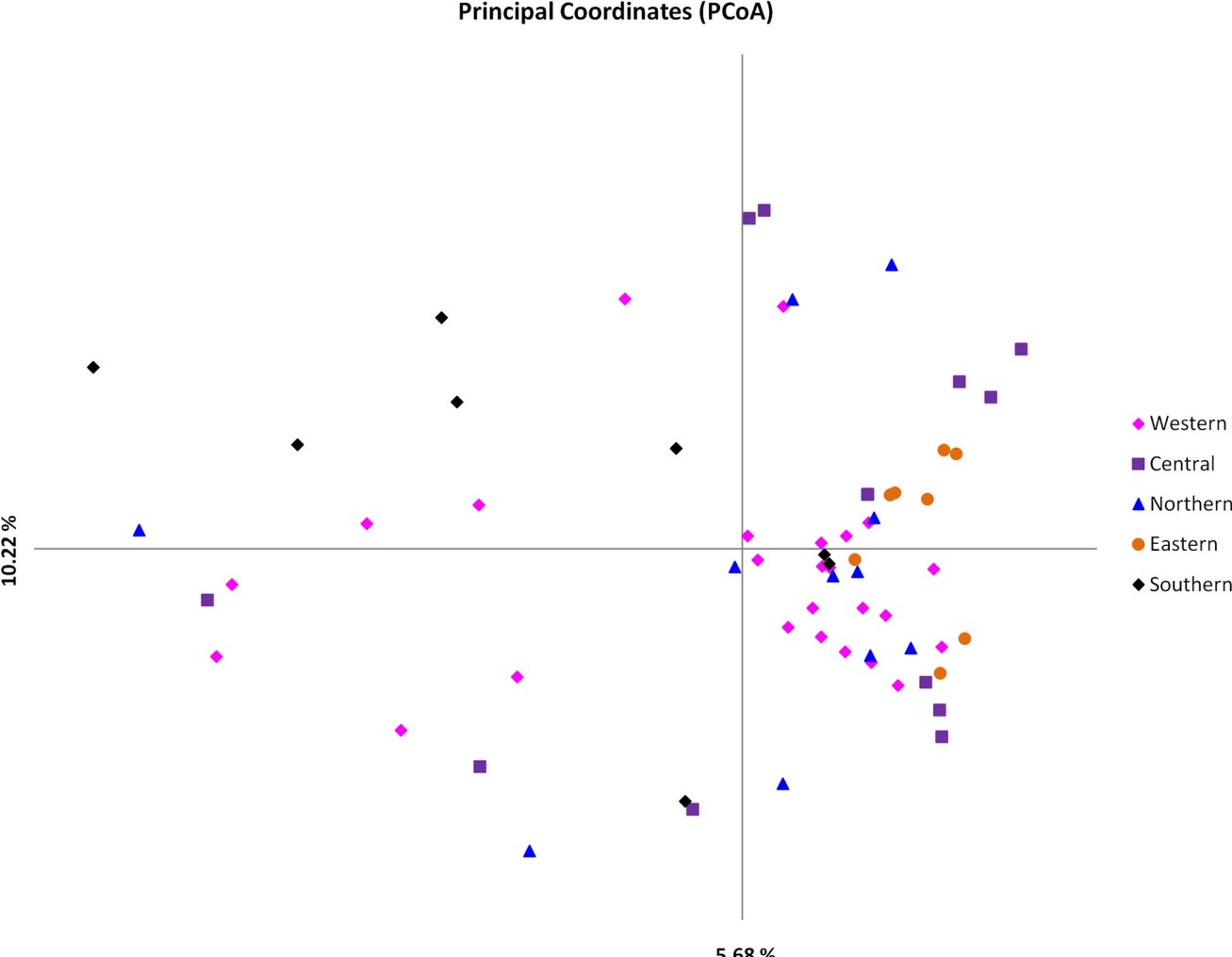

**Fig 9. Principal component analysis (PCA) of field samples from different regions of Burundi.** The samples are occupying all quadrants of the PCA and exhibiting low genetic differentiation across the different regions. The variation in proportion in the population data set given on each axis.

Eastern and Southern, the obtained index of association values of 0.5750, 0.1224 and 0.1647 coupled with $V_D$ values that were greater than the L values, indicated linkage disequilibrium and non-panmixia (non random mating) (Table 6). Overall, the $V_D$ value of 0.2407 that was greater than the L value of 0.1803 and the standard index of association of 0.0978, indicating linkage disequilibrium and non panmixia, were obtained when field samples from all regions under study and MC were treated as a single population (Table 6).

## Discussion

Theileriosis or East Coast fever is widely spread in east and central Africa. Acaricides as well as drugs are available for the control of the vector and parasite of ECF, respectively, but these tend to be costly for the local farmers and in most cases, immunization is the only cost effective and feasible method of ECF control. East Coast fever has been previously reported in Burundi [5, 41]. In the current study, the prevalence of *T. parva* in the different regions of Burundi was estimated from samples collected in 2014 using microscopic examination and ELISA. Using these methods, and in contrast to the previous study using samples collected in 2017 [41], 30%

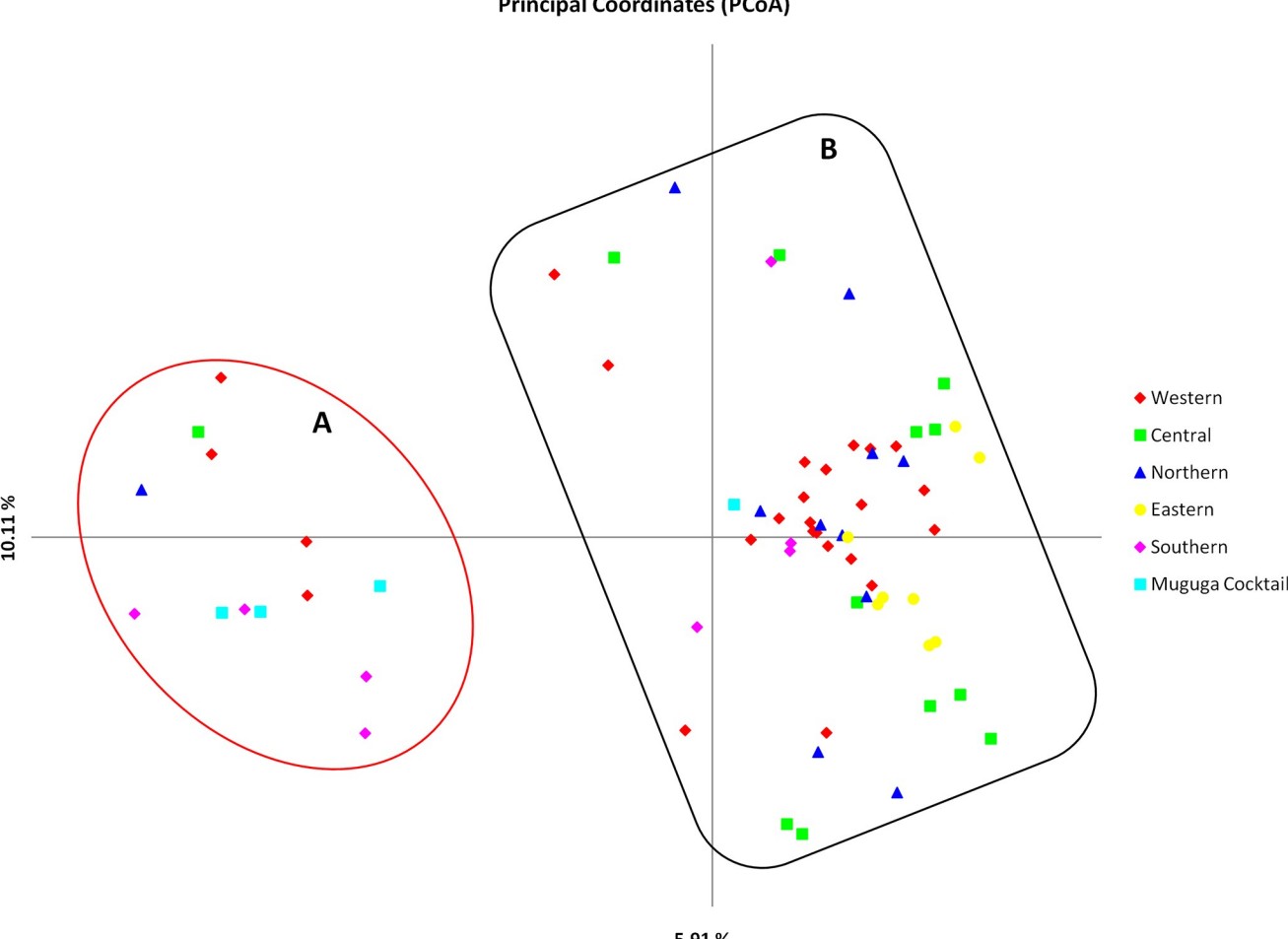

**Principal Coordinates (PCoA)**

**Fig 10. Principal component analysis of field samples from Burundi and Muguga cocktail vaccine.** Moderate sub-structuring between populations A (comprising of samples from the Northern, Southern, Western and Central regions as well as Muguga cocktail vaccine) and B (comprising of samples from the Northern, Southern, Eastern, Western and Central regions together with the Muguga cocktail vaccine) can be observed. The numbers in the parenthesis on each axis indicate the proportion of variance in the population data set.

(Table 1) and 60.1% (Table 2) of the samples were positive for *T. parva* (Tables 1 and 2), thus indicating that animals in these regions were carriers and had been exposed to *T. parva*. Furthermore, 79% of DNA samples screened by PCR using p104 primers were also positive for *T. parva* (Table 3). The low proportion of positive samples on microscopy can be attributed to the low sensitivity of this technique as a diagnostic tool. However, the higher proportion of *T. parva* positive antibodies in this region implies that a larger number of animals have been exposed to *T. parva* and further, the number of confirmed ECF positive samples on PCR indicates that there is also a higher proportion of carriers in Burundi. Overall, in the same manner as the previous study [41], this data shows the presence of *T. parva* and its carriers in the cattle population which ultimately pose a source of infection to naive unvaccinated animals in Burundi.

The presence of ECF is a serious impediment to the growth of the livestock sector in Burundi and as such, effective control and preventive measures such as ITM are warranted. However, before any immunization campaign can be made policy, characterization of the *T. parva* populations prevailing in a particular area is important for the assessment of the

## Principal Coordinates (PCoA)

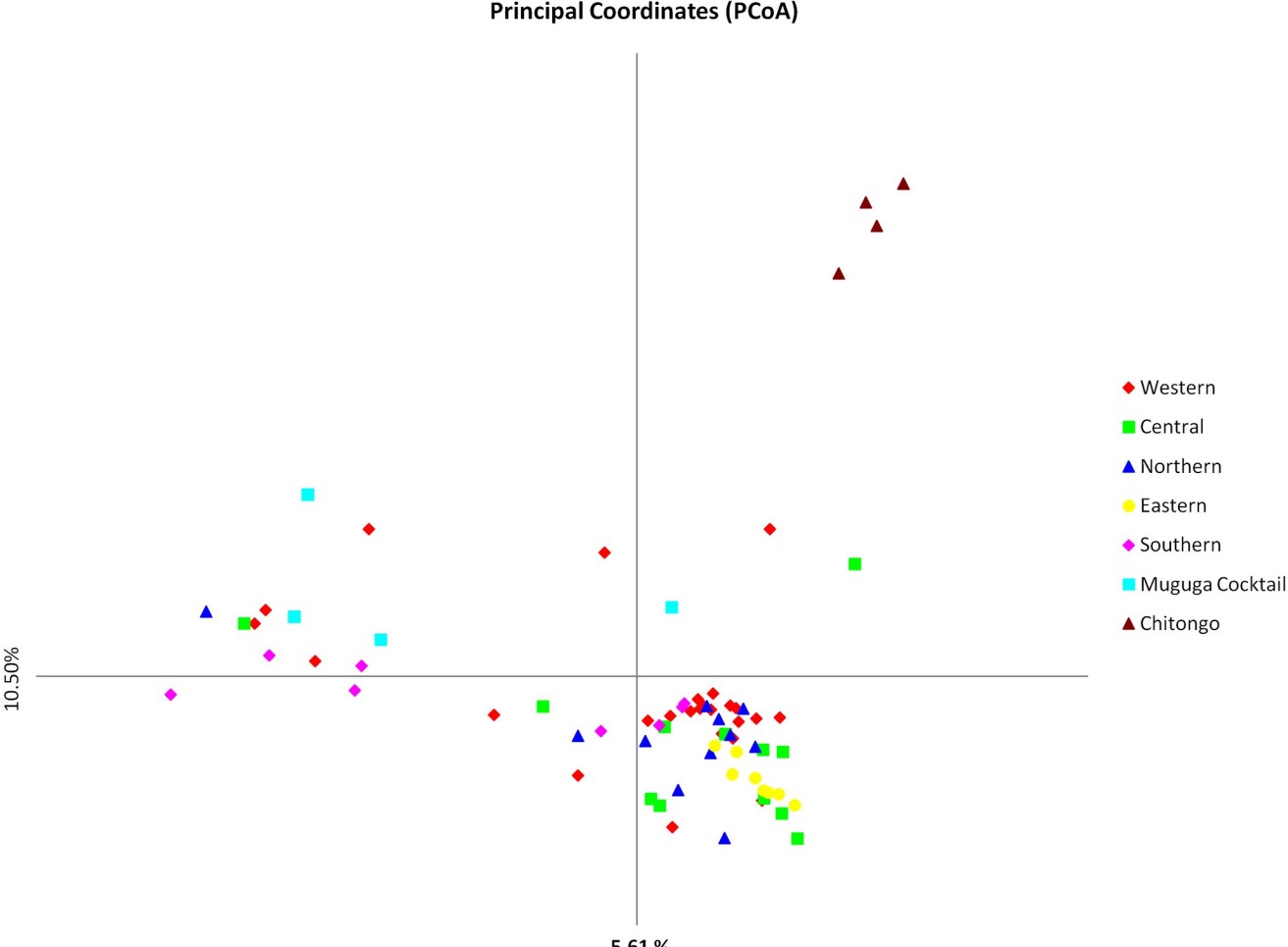

**Fig 11. Principal component analysis of Burundi field samples together with the Muguga cocktail and Chitongo vaccines.** The PCA plot indicates great genetic sub-structuring between the Chitongo vaccine and the Burundi field samples together with the Muguga cocktail vaccine. The variance in the data set of the population is given by the number in the parenthesis on each axis.

**Table 6. Genetic analyses of field samples from the Western, Central, Northern, Eastern and Southern regions of Burundi and the Muguga cocktail (Muguga, Kiambu and Serengeti as well as the MCL01) vaccine.**

| Population | N | $H_t$ | Genotypes/loci | $V_D$ | L | P—Value | $I_A^S$ | Linkage |
|---|---|---|---|---|---|---|---|---|
| **Western region** | 25 | 0.961 | 13.298 | 0.1832 | 0.2100 | 0.39 | -0.0024 | LE |
| **Central region** | 13 | 0.931 | 7.153 | 1.0604 | 0.411 | 0.01 | 0.5750 | LD |
| **Northern region** | 11 | 0.967 | 8.373 | 0.1764 | 0.2135 | 0.11 | 0.0303 | LE |
| **Eastern region*** | 8 | 0.821 | 3.703 | 1.0622 | 1.0622 | 0.06 | 0.1224 | |
| **Southern region*** | 8 | 0.879 | 4.643 | 0.8399 | 0.7659 | 0.02 | 0.1647 | LD |
| **All regions combined** | 65 | 0.912 | 7.434 | 0.2314 | 0.1765 | 0.01 | 0.0951 | LD |
| **Muguga Cocktail** | 6 | 0.707 | 2.745 | | | | | |
| **All regions plus Muguga cocktail combined** | 71 | 0.878 | 6.652 | 0.2407 | 0.1803 | 0.01 | 0.0978 | LD |

N: number of samples, $H_t$: Estimated Heterozygosity, $V_D$: mismatch variance (linkage analysis), L: upper 95% confidence limit of Monte Carlo simulation (linkage analysis) $I_A^S$: Standard Index of Association.

similarity of the field population of *T. parva* with that in the vaccine stocks. This approach has been used more recently in Rwanda [27]. On that basis, this study determined the prevalence of *T. parva* and further analyzed the sequence diversity of p67 as well as two *T. parva* CTL antigens Tp1 and Tp2. The study also assessed the population structure of *T. parva* in the Northern, Western, Central, Eastern and Southern regions of Burundi using a panel of 5 satellite markers.

Multiple nucleotide sequence alignment of the p67 gene showed a 129 bp deletion in the field sample sequences and while it was previously well accepted that this deletion was only present in cattle derived strains of *T. parva* and used to distinguish between cattle-derived and buffalo-derived strains [17], recent studies have shown that it is difficult to distinguish between cattle-derived and buffalo-derived *T. parva* strains using p67 [40, 42, 43]. Nevertheless, phylogenetic analysis of the p67 gene revealed that all the field sample sequences clustered with Muguga M67476 allele 1 and KY912962 allele 1 reference sequences (Fig 2). This implies that within the field samples investigated, the p67 allele 1, present in Muguga (M67476) was the only allele detected in Burundi and none of the other alleles are present (Fig 2). Owing to the lack of information regarding the diversity of p67 in buffaloes and cattle in Burundi, we can only speculate as to whether the *T. parva* detected in this study is cattle derived or not based on p67 gene. Overall, the detection of the same allele in both Muguga and the field samples indicates similarity between the field population and Muguga.

In comparison and contrast to other studies [18, 27, 44, 45], phylogenetic analysis of CTL Tp1 (Fig 3) and Tp2 (Fig 5) antigens revealed 5 and 4 clusters, respectively. On both phylogenetic trees, the majority of samples (n = 87 for Tp1 and n = 69 for Tp2) clustered with the Muguga, Kiambu and Serengeti isolates while the minority (n = 19 for Tp1 and n = 34 for Tp2) clustered with Chitongo vaccine isolate (Figs 3 and 5). This implies that the majority of field samples were closely related to Muguga, Kiambu and Serengeti as compared to Chitongo vaccine isolate. The remaining samples (n = 21 for Tp1 and n = 15 for Tp2) did not cluster with any of the vaccine isolates and may indicate the presence of a local population that is different but related to MC and the majority of the field samples. Further, the majority of Tp1 and Tp2 epitopes in field samples were 100% similar to those in Muguga, Kiambu and Serengeti vaccine isolates when compared to Chitongo vaccine isolate. In addition, the MJ networks for Tp1 (Fig 4) and Tp2 (Fig 6) both showed the presence of expanding populations [46] with the majority of haplotypes being related, but not similar, to MC. Furthermore, no evidence of segregation according to the geographical origin of the samples was observed on epitope sequence, phylogenetic or haplotype analysis, instead all epitopes and haplotypes were evenly distributed across the different regions under study. A low nucleotide diversity was also observed on Tp1, indicating conservation of the gene while Tp2 showed a high level of diversity, indicating high polymorphism as previously reported [18, 27, 44, 45] and as further evidenced by the Tp2 MJ network (Fig 6). Tp1 and Tp2 showed greater proportions of dS than dN with negative purifying selection. While Tp1 exhibited no positive selection sites indicative of an absence of selection pressure, Tp2 possessed 30 purifying selection sites and 8 positive/diversifying sites indicating that although the majority of the gene was under negative selection, there was some extent of selection pressure occurring on the gene either due to host factors or intervention strategies such as vector control or development of minor enzootic stability. Overall, on both genes, the genetic variation observed was within the population while the least variation was among the populations as evidenced by AMOVA.

In order to improve and consolidate the results of sequence diversity, population genetic analysis using 5 polymorphic markers of *T. parva* was conducted and revealed high gene diversities across all loci and populations. High estimated heterozygosities and mean number of genotypes per loci were also observed (Table 6). Field samples shared a very low number of

alleles with MC and none with Chitongo (Fig 8). Furthermore, a high proportion of unique alleles were also observed among the field sample populations and also when MC was included in the analysis. This indicates the presence of geographical sub-structuring among the field sample populations and also between the field sample populations and MC. This level of sub-structuring was visualized by PCA (Figs 9–11). With the exception of the Eastern region population whose samples were restricted to the upper and lower right quadrants, samples from all the other populations were distributed in all the quadrants. Introduction of the MC population in the PCA influenced genetic sub-structuring producing to minor populations designated population A and B. Further assessment of the sub-structuring observed on allele frequency and PCA using Wright's F index revealed low genetic differentiation ($F_{ST} = 0.047$) among the field populations, low to moderate genetic differentiation ($F_{ST} = 0.065$) between MC and the field population and moderate genetic differentiation between minor populations A and B ($F_{ST} = 0.114$). Further, LE and panmixia (random mating) was observed in the Western and Northern populations and LD and non-panmixia in the other populations (Table 6). The low genetic differentiation, LE and panmixia in the Western and Northern populations can be attributed to the interaction of the parasite populations within each region as a result of free movement and mixing of cattle, absence of effective control programs focused on both the vector and the parasite and absence of selection pressure due to the lack of immunization programs. Intra population LD and non panmixia in the other populations on the other hand could be because samples originated from different parts or towns or villages within a particular region and of course had no chance to interact. However, a more probable reason for the LD and non-panmixia could be attributed simply to the low sample size as was observed in other studies [27–29]. Overall, the field populations with or without MC population exhibited LD and non-panmixia and this might be due to the low number of markers used in the study, distances between the different regions which impedes the free movement of cattle and vectors [29], and low tick challenge due to the presence of control strategies such as acaricide control in some regions which ultimately prevents the interaction of different parasite populations from different regions [28]. Thus, population genetic analysis showed that even though some level of genetic differentiation exists among the field sample populations, it occurs at a low level ($F_{ST} = 0.047$) and the different populations from the different regions are highly similar. It also showed that these populations do not share a lot of alleles with MC. Furthermore, the low to moderate differentiation between the field populations and MC indicates similarity between the field samples and the MC vaccine components and the sub-structuring observed on PCA could be as a result of the lack of interaction between the field sample populations and MC in the field since MC is a laboratory stock. Nevertheless, in a similar manner as sequence diversity of p67, Tp1 and Tp2, the population genetic data has demonstrated some level of support for the possible use of MC vaccine in field trial challenges. Thus vaccination with a combination of Muguga, Kiambu and Serengeti in the different regions of Burundi is likely to produce desirable protection against challenge from the field especially if the vaccinated cattle also exhibit MHC class I genes or haplotypes [21, 47].

## Conclusions

The p67, Tp1 and Tp2 sequence diversity data revealed the presence of a population whose individuals are closely related to and share similar epitopes with MC components. This was further supported by the microsatellite analysis of the same field samples using five satellite markers which also revealed a close similarity among the field samples from different regions of Burundi and also to the MC components. Therefore, the careful use of MC vaccine in field challenge trials in the different regions of Burundi is likely to confer protection against ECF as

previously demonstrated in neighboring Rwanda where a similar approach produced a protection level of 81.7% in calves [27]. However, a field challenge trial could not be conducted in this study.

## Supporting information

**S1 Fig. Allele frequencies from field samples from Burundi.** The allele frequencies from Burundi field samples in this study showing the presence of both shared and unique alleles. MS39 shows the highest number of unique alleles while MS19 shows the least. A total of 4 shared alleles is observed on each loci.
(TIF)

**S2 Fig. Allele frequencies from field samples from Burundi.** The allele frequencies from Burundi field samples in this study showing the presence of both shared and unique alleles. MS39 shows the highest number of unique alleles while MS19 shows the least. A total of 4 shared alleles is observed on each loci.
(TIF)

**S3 Fig. Allele frequencies from Burundi field samples, Muguga cocktail and Chitongo vaccine stocks.** Muguga cocktail shares only two alleles with Burundi field samples on loci MS19 and ms9 while none are shared on the remaining loci. Chitongo vaccine stock does not share any alleles with the Burundi field samples.
(TIF)

**S4 Fig. Allele frequencies from Burundi field samples, Muguga cocktail and Chitongo vaccine stocks.** Muguga cocktail shares only two alleles with Burundi field samples on loci MS19 and ms9 while none are shared on the remaining loci. Chitongo vaccine stock does not share any alleles with the Burundi field samples.
(TIF)

**S1 Table. Theileria parva satellite markers used in this study to genotype Burundi field samples, Muguga cocktail and Chitongo vaccines stocks.**
(DOCX)

**S2 Table. Summarized list of field sample sequences with 100% nucleotide homology used for the generation of phylogenetic trees for the Tp1 and Tp2 genes.**
(DOCX)

**S3 Table. Description of Burundi field sample sequences used in this study.**
(XLSX)

## Acknowledgments

We thank the staff of the National Veterinary Research Laboratory, Directorate of Animal Health, Burundi for their role in sample collection and parasitology analysis. We also thank Mr. Osbert Mphukira Pangani and other technical staff of the Centre for Ticks and Tick-Borne Diseases as well as the technical staff at the University of Zambia, School of Veterinary Medicine for some of the laboratory work. We are grateful to Phil Toye and Roger Pelle of the International Livestock Research Institute for the helpful advice in the design of these experiments.

## Author Contributions

**Conceptualization:** David Kalenzi Atuhaire, Walter Muleya, Victor Mbao, Jeremy Salt, Antony Jim Musoke.

**Data curation:** Walter Muleya, Boniface Namangala, Antony Jim Musoke.

**Formal analysis:** David Kalenzi Atuhaire, Walter Muleya, Boniface Namangala.

**Funding acquisition:** David Kalenzi Atuhaire, Walter Muleya, Victor Mbao, Jeremy Salt, Antony Jim Musoke.

**Investigation:** David Kalenzi Atuhaire, Walter Muleya, Jeremy Salt, Antony Jim Musoke.

**Methodology:** David Kalenzi Atuhaire, Walter Muleya, Joseph Niyongabo, Lionel Nyabongo, Deogratias Nsanganiyumwami, Antony Jim Musoke.

**Project administration:** David Kalenzi Atuhaire, Victor Mbao, Jeremy Salt, Antony Jim Musoke.

**Resources:** Walter Muleya, Lionel Nyabongo, Deogratias Nsanganiyumwami, Jeremy Salt.

**Software:** Walter Muleya.

**Supervision:** David Kalenzi Atuhaire, Walter Muleya, Jeremy Salt, Boniface Namangala, Antony Jim Musoke.

**Validation:** David Kalenzi Atuhaire, Walter Muleya, Victor Mbao, Jeremy Salt.

**Visualization:** David Kalenzi Atuhaire, Walter Muleya.

**Writing – original draft:** David Kalenzi Atuhaire, Walter Muleya, Victor Mbao, Joseph Niyongabo, Lionel Nyabongo, Deogratias Nsanganiyumwami, Jeremy Salt, Boniface Namangala, Antony Jim Musoke.

**Writing – review & editing:** David Kalenzi Atuhaire, Walter Muleya, Victor Mbao, Joseph Niyongabo, Lionel Nyabongo, Deogratias Nsanganiyumwami, Jeremy Salt, Boniface Namangala, Antony Jim Musoke.

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
