## [Decision Letter · Decision Letter 0]

30 Mar 2021

PONE-D-21-01358

Molecular characterization and population genetics of Theileria parva in Burundi’s unvaccinated cattle: Towards the introduction of East Coast Fever vaccine

PLOS ONE

Dear Dr. Muleya,

Thank you for submitting your manuscript to PLOS ONE. After careful consideration, we feel that it has merit and is nearly ready for publication in  PLOS ONE. Therefore, we invite you to submit a revised version of the manuscript that addresses the points raised during the review process.

We look forward to receiving your revised manuscript.

Kind regards,

Benjamin M. Rosenthal

Academic Editor

PLOS ONE

Journal Requirements:

4. Thank you for stating the following in the Funding Section of your manuscript:

[This publication has been supported by the Global Alliance for Livestock and Veterinary 843medicines (GALVmed) with funding from the Bill & Melinda Gates Foundation and UK aid 844from the UK Government.]

 [Financial support for this work was within the framework of the project on the characterization and population genetics of Theileria parva strains in eastern, central and southern Africa, grant number UOZ-R34A0548A3 received by AJM from the Global alliance for livestock medicines.   The funders had no role in study design, data collection and analysis, decision to publish, or preparation of the manuscript.]

6. We note that Figure 1 in your submission contain [map/satellite] images which may be copyrighted. All PLOS content is published under the Creative Commons Attribution License (CC BY 4.0), which means that the manuscript, images, and Supporting Information files will be freely available online, and any third party is permitted to access, download, copy, distribute, and use these materials in any way, even commercially, with proper attribution. For these reasons, we cannot publish previously copyrighted maps or satellite images created using proprietary data, such as Google software (Google Maps, Street View, and Earth). For more information, see our copyright guidelines: http://journals.plos.org/plosone/s/licenses-and-copyright.

You may seek permission from the original copyright holder of Figure 1 to publish the content specifically under the CC BY 4.0 license. 

If you are unable to obtain permission from the original copyright holder to publish these figures under the CC BY 4.0 license or if the copyright holder’s requirements are incompatible with the CC BY 4.0 license, please either i) remove the figure or ii) supply a replacement figure that complies with the CC BY 4.0 license. Please check copyright information on all replacement figures and update the figure caption with source information. If applicable, please specify in the figure caption text when a figure is similar but not identical to the original image and is therefore for illustrative purposes only.

Reviewers' comments:

Reviewer's Responses to Questions

**Comments to the Author**

1. Is the manuscript technically sound, and do the data support the conclusions?

Reviewer #1: Yes

Reviewer #2: Yes

2. Has the statistical analysis been performed appropriately and rigorously? 

Reviewer #1: Yes

Reviewer #2: Yes

3. Have the authors made all data underlying the findings in their manuscript fully available?

Reviewer #1: Yes

Reviewer #2: Yes

4. Is the manuscript presented in an intelligible fashion and written in standard English?

Reviewer #1: Yes

Reviewer #2: Yes

5. Review Comments to the Author

Reviewer #1: The investigators conducted an extensive survey to determine the prevalence and composition of T. parva variants in Burundi Africa, to determine the potential value of introducing a live vaccine cocktail to control East Coast Fever. Although the sample size for the different regions of Burundi, the samples were sufficient to obtain an estimate of the genetic composition of the T. parva variants in Burundi. Importantly, the results revealed the genetic/antigenic profile of the T. parva variants present in Burundi were similar to the variants that make up the Maguga vaccine cocktail. The data indicate the introduction and use of the Maguga cocktail to control ECF in Burundi is warranted. It is recognized that there is a need for continuation of efforts to develop a vaccine that can replace the infect and treat method of vaccination.

Reviewer #2: The authors investigated the prevalence and population genetic structure of Theileria parva in cattle from Burundi. Prevalence was determined using microscopy, ELISA using the PIM antigen and PCR targeting the p104 gene. Prevalence on both ELISA and PCR ranged from 60-64%. Only one p67 allele was present with the characteristic 129 bp deletion associated with cattle-derived T. parva and phylogenetic analysis indicated that all sequences clustered with the Muguga M67476 allele. Analysis of the CTL Tp1 and Tp2 epitopes indicate sharing of epitoptes with the Muguga cocktail and this was confirmed with phylogenetic analysis that showed that these genes cluster with Muguga, Kiambu and Serengeti isolates or with Chitongo. A small number of sequences did not cluster with the vaccine isolates, suggesting that this region do contain novel strains. No geographic sub-structuring were observed for the Tp1 and Tp2 sequences. Micro- and mini-satellite analysis indicated that few alleles were shared with the vaccine isolates at this level, with geographic sub-structuring present. Low to medium genetic diversity was observed in the populations, with linkage equilibrium and panmixia observed in Northern and Western populations, while non-panmixia and lingage disequilibrium was observed for other populations. Reasons for these differences are discussed. The authors conclude that enough similarity exist between the Burundi strains and vaccine isolates to warrant deployment of vaccination with the Muguga cocktail since cross-protection is probable based on the data generated. They are most probably correct in this assessment. The study comprise an in-depth analysis of the genetic diversity of T. parva in Burundi and will serve as a gold standard for this region. It is clearly written and the conclusions are supported by the results. A few minor issues follow.

Page 12, lines 259-267: A table similar to Table 2 should be added for the PCR results.

Page 14, line 298: Replace “homology” with “identity” since homology refers to relationship by vertical descent and not a percentage sequence identity or similarity.

Page 14, line 304: … MC epitopes …

Page 15, line 327: See comment above for the use of homology and identity.

Page 16, line 335: It seems as if the median-joining network analysis method is not described in the methods section.

Page 16, line 349: Furthermore, similar ….

Page 17, line 366: See comment above for the use of homology and identity.

Page 17, line 372: See comment above for the use of homology and identity.

Page 27, line 545: … East Coast fever …

Page 29, line 582: … p67 in buffaloes …

References: Please check format of references.

Some references do not have page numbers.

Some references have DOI links and others not.

Species names needs to be italicized.

6. PLOS authors have the option to publish the peer review history of their article (what does this mean?). If published, this will include your full peer review and any attached files.

Reviewer #1: **Yes: **William C. Davis

Reviewer #2: No

---

## [Author Response · Author response to Decision Letter 0]

24 Apr 2021

Reponses to Reviewers

Issues raised by Editor and reviewers Authors response

Editor 

Please ensure that your manuscript meets PLOS ONE's style requirements, including those for file naming

 The authors agree with the editor and have since formatted the manuscript as advised. 

Please review your reference list to ensure that it is complete and correct. If you have cited papers that have been retracted, please include the rationale for doing so in the manuscript text, or remove these references and replace them with relevant current references. Any changes to the reference list should be mentioned in the rebuttal letter that accompanies your revised manuscript. If you need to cite a retracted article, indicate the article’s retracted status in the References list and also include a citation and full reference for the retraction notice. The authors agree with the editor and have reviewed and corrected the reference list. 

Please provide additional details regarding participant consent. In the ethics statement in the Methods and online submission information, please ensure that you have specified what type you obtained (for instance, written or verbal, and if verbal, how it was documented and witnessed). If your study included minors, state whether you obtained consent from parents or guardians. If the need for consent was waived by the ethics committee, please include this information. The authors agree with the editor and have edited the ethics statement to read as follows, "Ethical clearance for the recombinant DNA experiments in this study was approved by the University of Zambia Biomedical Research Ethics Committee (UNZABREC) under REF. 233-2019. Clearance for collection of samples was obtained from the Directorate of Animal Health, Burundi and verbal consent was sought from farmers. Blood sampling was done by a veterinarian from cattle owned by consenting farmers and did not involve endangered or protected animal species. The animals were handled humanely during sample collection". 

 [Financial support for this work was within the framework of the project on the characterization and population genetics of Theileria parva strains in eastern, central and southern Africa, grant number UOZ-R34A0548A3 received by AJM from the Global alliance for livestock medicines. The funders had no role in study design, data collection and analysis, decision to publish, or preparation of the manuscript.]

Please include your amended statements within your cover letter; we will change the online submission form on your behalf. The authors agree with the editor and have deleted the funding information in the main text of the manuscript. Further the statement "Financial support for this work was within the framework of the project on the characterization and population genetics of Theileria parva strains in eastern, central and southern Africa, grant number UOZ-R34A0548A3 received from the Global alliance for livestock medicines (GALVmed). The funders had no role in study design, data collection and analysis, decision to publish, or preparation of the manuscript", has been included in the cover letter for onward inclusion on the online submission form.

We note that you have included the phrase “data not shown” in your manuscript. Unfortunately, this does not meet our data sharing requirements. PLOS does not permit references to inaccessible data. We require that authors provide all relevant data within the paper, Supporting Information files, or in an acceptable, public repository. Please add a citation to support this phrase or upload the data that corresponds with these findings to a stable repository (such as Figshare or Dryad) and provide and URLs, DOIs, or accession numbers that may be used to access these data. Or, if the data are not a core part of the research being presented in your study, we ask that you remove the phrase that refers to these data The authors agree with the editor and have deleted the phrase "data not shown". In addition the statement " Multiple sequence alignment of the p67 gene nucleotide sequences generated in this study together with p67 reference sequences representing all four alleles of p67 showed the presence of a 129 bp deletion in all the field sample sequences (data not shown)" has also been deleted because the information in this statement is similar to the statements "phylogenetic analysis showed four clusters namely A, B, C and D (Fig 2). Within these clusters (Fig 2), sequences clustered according to the p67 allele types as previously described (41). All the field sample sequences formed a cluster with Muguga M67476 allele 1 and KY912962 allele 1 reference sequences in cluster A. None of the field samples clustered with any of the remaining alleles thus within the samples analysed, only allele type 1 which is present in Muguga was represented."

We note that Figure 1 in your submission contain [map/satellite] images which may be copyrighted. All PLOS content is published under the Creative Commons Attribution License (CC BY 4.0), which means that the manuscript, images, and Supporting Information files will be freely available online, and any third party is permitted to access, download, copy, distribute, and use these materials in any way, even commercially, with proper attribution. For these reasons, we cannot publish previously copyrighted maps or satellite images created using proprietary data, such as Google software (Google Maps, Street View, and Earth). For more information, see our copyright guidelines: http://journals.plos.org/plosone/s/licenses-and-copyright.

The authors respond by stating that the map presented as figure 1 does not possess any copyright issues because it was not obtained from anyone. Instead the map in this study was drawn using the opens source software QGIS 3.16.5 'Hannover' (https://qgis.org/en/site/forusers/download.html) and all the vector files were obtained from ICPAC Geoportal (http://geoportal.icpac.net/layers/geonode%3Aafr_g2014_2013_0), another open-source platform for creating and sharing geospatial data and maps. 

 Reviewer #1 

The investigators conducted an extensive survey to determine the prevalence and composition of T. parva variants in Burundi Africa, to determine the potential value of introducing a live vaccine cocktail to control East Coast Fever. Although the sample size for the different regions of Burundi, the samples were sufficient to obtain an estimate of the genetic composition of the T. parva variants in Burundi. Importantly, the results revealed the genetic/antigenic profile of the T. parva variants present in Burundi were similar to the variants that make up the Maguga vaccine cocktail. The data indicate the introduction and use of the Maguga cocktail to control ECF in Burundi is warranted. It is recognized that there is a need for continuation of efforts to develop a vaccine that can replace the infect and treat method of vaccination. The authors are very grateful for the comment from the reviewer and agree with the reviewer's views.

Reviewer #2 

The authors investigated the prevalence and population genetic structure of Theileria parva in cattle from Burundi. Prevalence was determined using microscopy, ELISA using the PIM antigen and PCR targeting the p104 gene. Prevalence on both ELISA and PCR ranged from 60-64%. Only one p67 allele was present with the characteristic 129 bp deletion associated with cattle-derived T. parva and phylogenetic analysis indicated that all sequences clustered with the Muguga M67476 allele. Analysis of the CTL Tp1 and Tp2 epitopes indicate sharing of epitoptes with the Muguga cocktail and this was confirmed with phylogenetic analysis that showed that these genes cluster with Muguga, Kiambu and Serengeti isolates or with Chitongo. A small number of sequences did not cluster with the vaccine isolates, suggesting that this region do contain novel strains. No geographic sub-structuring were observed for the Tp1 and Tp2 sequences. Micro- and mini-satellite analysis indicated that few alleles were shared with the vaccine isolates at this level, with geographic sub-structuring present. Low to medium genetic diversity was observed in the populations, with linkage equilibrium and panmixia observed in Northern and Western populations, while non-panmixia and lingage disequilibrium was observed for other populations. Reasons for these differences are discussed. The authors conclude that enough similarity exist between the Burundi strains and vaccine isolates to warrant deployment of vaccination with the Muguga cocktail since cross-protection is probable based on the data generated. They are most probably correct in this assessment. The study comprise an in-depth analysis of the genetic diversity of T. parva in Burundi and will serve as a gold standard for this region. It is clearly written and the conclusions are supported by the results The authors are grateful for the comment from the reviewer and agree with the views expressed. 

Page 12, lines 259-267: A table similar to Table 2 should be added for the PCR results. Table 3, similar to Table 2, has been inserted and the rest of the tables realigned. 

Page 14, line 298: Replace “homology” with “identity” since homology refers to relationship by vertical descent and not a percentage sequence identity or similarity. The authors agree with reviewer and have replaced "homology" with "identity"

Page 14, line 304: … MC epitopes … The authors agree with reviewer and have edited MC epitope to "MC epitopes".

Page 15, line 327: See comment above for the use of homology and identity The authors agree with reviewer and have replaced "homology" with "identity"

Page 16, line 335: It seems as if the median-joining network analysis method is not described in the methods section. The authors agree with the reviewer and have inserted the statement "Network ver. 10 was utilized to assess the haplotype similarities between the Tp1 and Tp2 nucleotide sequences of the vaccine stocks and the field samples (http://fluxus-engineering.com/)" in the data analysis section.

Page 16, line 349: Furthermore, similar … Furthermore, Similar … has been replaced with "Furthermore, similar …"

Page 17, line 366: See comment above for the use of homology and identity. The authors agree with reviewer and have replaced "homology" with "identity"

Page 17, line 372: See comment above for the use of homology and identity. The authors agree with reviewer and have replaced "homology" with "identity"

Page 27, line 545: … East Coast fever … "East coast fever …" edited to "East Coast fever"

Page 29, line 582: … p67 in buffaloes … "p67 in Buffaloes" edited to "p67 in buffaloes"

References: Please check format of references. Some references do not have page numbers.

Some references have DOI links and others not.

Species names needs to be italicized. All the references have been checked and conform to journal requirements

---

## [Editor Report · Decision Letter 1]

28 Apr 2021

Molecular characterization and population genetics of Theileria parva in Burundi’s unvaccinated cattle: Towards the introduction of East Coast Fever vaccine

PONE-D-21-01358R1

Dear Dr. Muleya,

We’re pleased to inform you that your manuscript has been judged scientifically suitable for publication and will be formally accepted for publication once it meets all outstanding technical requirements.

Kind regards,

Benjamin M. Rosenthal

Academic Editor

PLOS ONE
---

## [Editor Report · Acceptance letter]

5 May 2021

PONE-D-21-01358R1 

Molecular characterization and population genetics of *Theileria parva* in Burundi’s unvaccinated cattle: Towards the introduction of East Coast Fever vaccine 

Dear Dr. Muleya:

I'm pleased to inform you that your manuscript has been deemed suitable for publication in PLOS ONE. Congratulations! Your manuscript is now with our production department. 

Kind regards, 

on behalf of

Dr. Benjamin M. Rosenthal 

Academic Editor

PLOS ONE